

# Parameterizing anisotropic reflectance of snow surfaces from airborne digital camera observations in Antarctica

Tim Carlsen[1,*], Gerit Birnbaum[2], André Ehrlich[1], Veit Helm[2], Evelyn Jäkel[1], Michael Schäfer[1], and Manfred Wendisch[1]

[1]Leipzig Institute for Meteorology, Leipzig University, Leipzig, Germany
[2]Alfred Wegener Institute, Helmholtz Centre for Polar and Marine Research, Bremerhaven, Germany
[*]Now at: Department of Geosciences, University of Oslo, Oslo, Norway

**Correspondence:** Tim Carlsen (tim.carlsen@geo.uio.no)

**Abstract.** The angular reflection of solar radiation by snow surfaces comprises an important boundary condition for radiative transfer simulations. In polar regions, the surface reflection is particularly anisotropic due to low sun elevations and the highly anisotropic scattering phase function of ice crystals. This anisotropy needs to be considered in the angular modeling of the surface reflection, which is essential for satellite remote sensing techniques. To quantify the snow reflection properties, the

hemispherical-directional reflectance function (HDRF) of snow surfaces was derived from airborne measurements using a digital 180° fish-eye camera (green channel, 490-585 nm wavelength band) in Antarctica during austral summer in 2013/14. This function was measured for different surface roughnesses, optical-equivalent snow grain sizes, and solar zenith angles. The airborne observations covered an area of around $1000 \times 1000 \, \mathrm{km}^2$ in the vicinity of Kohnen station (75°0′ S, 0°4′ E) at the outer part of the East Antarctic Plateau. The observations over Dronning Maud Land include regions with higher (coastal areas)

and lower (inner Antarctica) precipitation amounts and frequencies. The digital camera was calibrated in terms of spectral radiances and installed in a downward-looking configuration on a research aircraft. It provides upward, angular-dependent radiance measurements from the lower hemisphere. The comparison of HDRF data derived for smooth and rough snow surfaces (sastrugi) showed significant differences, which are superimposed on the diurnal cycle. By inverting a semi-empirical, kernel-driven bidirectional reflectance distribution function (BRDF) model, the measured HDRF of snow surfaces was parameterized

with respect to solar zenith angle, surface roughness, and optical-equivalent snow grain size. This allows a direct comparison of the HDRF measurements with the BRDF derived from the Moderate Resolution Imaging Spectroradiometer (MODIS) satellite product MCD43. For the analyzed cases, MODIS observations generally underestimated the anisotropy of the surface reflection. Some of these deviations between airborne and MODIS satellite retrievals are likely linked to short-term changes in snow properties.

## 1 Introduction

Surface reflection in the polar regions plays an important role in the Earth's climate system. The snow surface albedo is highly variable both on a temporal and spatial scale. This causes uncertainties in determining the surface solar radiative energy budget in these areas (e.g., Kuipers Munneke et al., 2008). As such it is essential to monitor the reflective properties of snow surfaces



for accurately predicting future climate change in the polar regions. However, their remote and harsh environment makes them difficult to access and requires remote sensing techniques to observe snow surface reflection and its influencing parameters.

The high spatial and temporal variability of the snow surface albedo necessitates continuous observations with global coverage, which are provided by satellite instruments. Satellites monitor the reflectance (i.e., reflected radiance in units of W m$^{-2}$ sr$^{-1}$) at the top of the atmosphere (TOA). However, they are restricted in terms of the number of available observation angles and spectral bands as well as their temporal resolution. The processing of satellite measurements for monitoring the broadband surface albedo typically requires three steps (e.g., Qu et al., 2015): Atmospheric correction, modeling of the angular reflectance, and narrow-to-broadband conversion. During the first step, the TOA reflectance is converted into a surface reflectance by means of an atmospheric radiative transfer parameterization (e.g., Vermote and Kotchenova, 2008). For the calculation of the narrowband surface albedo, an accurate knowledge of the bidirectional reflectance distribution function (BRDF) of the surface is required due to the limited number of available observation angles. The BRDF describes the scattering of electromagnetic (EM) radiation from one incident direction into another direction of the hemisphere. By using a linear combination of the discrete spectral band measurements with specific weighting, the broadband surface albedo is calculated (e.g., Brest and Goward, 1987; Klein and Stroeve, 2002; Liang et al., 2003; Pohl et al., 2020). The largest uncertainty in this three-step process is introduced by the angular dependence of the surface BRDF, especially when the reflection of the underlying surface is highly anisotropic. This is the case for the reflection of solar radiation at snow surfaces in polar regions due to low sun elevations and the anisotropic scattering phase function of snow crystals. As the snow surface of the Antarctic ice sheet is also used for the validation and cross-calibration of polar orbiting satellites (Jaross and Warner, 2008), a thorough understanding of the anisotropic reflection is needed. However, as an infinitesimal quantity, the BRDF of a surface cannot directly be measured under natural illumination conditions. From a rigorous physical point of view, most satellite, airborne, and ground-based instruments measure the hemispherical-directional reflectance factor (HDRF, Schaepman-Strub et al., 2006) if the reflectance is constant over the full cone angle of the instrument's field of view (FOV). In contrast to the BRDF, the HDRF includes incident irradiance from the entire hemisphere. An effective BRDF can only be derived from HDRF observations if the atmospheric influence is considered and the FOV is sufficiently small (e.g., Gatebe et al., 2003; Lyapustin et al., 2010).

For example, the retrieval of the aerosol optical depth (AOD) over highly reflecting surfaces is complicated, as it is hard to separate the surface and aerosol contributions to the measured reflectance at the TOA (Tomasi et al., 2015), especially for the comparably low values of AOD in polar regions (Mei et al., 2013). Due to its high surface albedo, most of the solar radiation incident on a snow surface is reflected, which depends strongly on the BRDF of the snow surface and, thus, on the illumination and viewing geometry. In general, most of the photons are scattered by the snow grains into the forward scattering direction. Yang et al. (2014) compared Moderate Resolution Imaging Spectroradiometer (MODIS) measurements of AOD at 550 nm over Eastern China with ground-based data from the Aerosol Robotic Network (AERONET). Using a Lambertian model, 54 % of the regression points fell within the expected error envelope around the 1:1 line. This result is improved to 69 % if a more sophisticated, non-Lambertian model accounting for anisotropic reflection at the land surface is applied.

Yet, the BRDF varies with snow grain morphology, the solar zenith and azimuth angles, the liquid water content of the snowpack, the snow impurity type and concentration, the dimension and orientation of surface roughness structures, and the





wavelength. To account for these effects, an accurate model of the snow BRDF is needed. A BRDF model can be either physical, empirical, or semi-empirical. Physical BRDF models (e.g., Cook and Torrance, 1982) accurately simulate the scattering of an EM wave at a surface by applying physical laws. The high accuracy comes at the cost of very complex computations. Empirical BRDF models (e.g., Phong, 1975; Walthall et al., 1985) mimic the surface reflection by means of a simple, non-physical

formulation. However, the drawback of the rather simple computations is their restricted accuracy. Semi-empirical models (e.g., Martonchik et al., 1998; Lucht et al., 2000) use simple, direct parameterizations of a more complex physical BRDF with a limited number of independent parameters.

The anisotropic reflection by snow surfaces was investigated by Li (1982) with simulations of the snow BRDF using Mie theory and the doubling and adding method. Applying Mie theory and the discrete ordinate method, Han (1996) retrieved the

surface albedo from satellite measurements in the Arctic. Leroux and Fily (1998) developed a snow BRDF model including the effect of sastrugi by means of regularly spaced, identical, and rectangular protrusions. Leroux et al. (1998) and Leroux et al. (1999) employed the doubling and adding method, Mie theory, and ray tracing to develop a polarized BRDF model. Comparing the simulated values with observations in the principal plane, they found that the BRDF in the near-infrared wavelength range is strongly affected by the snow grain shape, whereby simulations assuming hexagonal particle shapes yield an improved

agreement with the observations compared to assuming spheres. Accordingly, Aoki et al. (2000) stressed the importance of the particle shape assumption for the scattering phase function used in their BRDF model (applying both Mie theory and the Henyey-Greenstein phase function) when comparing to observations between 0.52 μm and 2.21 μm wavelength. Kokhanovsky and Zege (2004) demonstrated the use of an asymptotic analytical equation to model the reflectance of snow. Their approach represents the snow grains as fractal particles and, thus, accounts for their non-sphericity. Comparing this asymptotic model to

in situ observations of the BRDF, Kokhanovsky et al. (2005) found a generally good agreement but a reduced model accuracy in the solar principal plane at large observation angles. In contrast to considering snow grains as independent scatterers of fractal shape, Malinka (2014) provided a framework to calculate the inherent optical properties applying the concept of stereology that considers snow as a random mixture of ice and air. Malinka et al. (2016) combined the stereological approach with analytical, asymptotic formulas to calculate the bidirectional reflectance. Their model showed a high accuracy when compared to albedo

observations of snow-covered sea ice.

The comparison of in situ measured BRDF with simulations is essential in terms of model validation. Observations of the BRDF or HDRF are conducted using a variety of different measurement concepts. For ground-based applications, manual or automated gonio-spectrometer systems are employed (e.g., Painter et al., 2003; Pegrum et al., 2006; Bourgeois et al., 2006a). Kuhn (1985) observed a peak in reflectance in the azimuthal directions up to 60° to both sides of the solar azimuth that

becomes more prominent with increasing solar zenith angle and snow grain size. Marks et al. (2015) measured the HDRF of snow surfaces between 400 and 1600 nm wavelength with the Gonio Radiometric Spectrometer System (GRASS, Pegrum et al., 2006) at Dome C, Antarctica. Their observations also showed enhanced forward scattering. In addition, they observed a larger anisotropy of the surface reflection at longer wavelengths. Employing a gonio-spectrometer, Bourgeois et al. (2006b) measured strong variations of the HDRF between 0.6 and 13 (in the wavelength range 350-1050 nm) depending on the solar

zenith angle and the surface roughness at Summit, Greenland. Measurements with the Automated Spectro-Goniometer (ASG,





Painter et al., 2003) showed a decrease of snow HDRF at all wavelengths between 400 to 2500 nm when the snow grain size increased from 80 to 280 μm (Painter and Dozier, 2004b). Further comparisons with the results of a forward discrete ordinates radiative transfer model (Stamnes et al., 1988) revealed larger deviations between the simulations and observations for more complex crystals. The importance of the crystal habit for the anisotropy of reflection at snow surfaces was further emphasized

by Dumont et al. (2010) and Stanton et al. (2016). The latter measured increasing anisotropy of the surface reflection during the growth of surface hoar in the laboratory. Several studies observed systematically less anisotropy for a typical snow BRDF than estimated from simulations (Warren et al., 1998; Painter and Dozier, 2004a; Hudson et al., 2006; Hudson and Warren, 2007). In the solar principal plane, the models mainly overestimate the forward scattering and underestimate the backward scattering. Implementing non-spherical grains in the BRDF models (e.g., Mishchenko et al., 1999; Xie et al., 2006; Jin et al., 2008)

improves the comparison with observations. The nonspherical model of Jin et al. (2008) agrees within ± 10 % for viewing zenith angles less than 60° with observations in Antarctica performed by Hudson et al. (2006). However, the asymmetry between forward and backward scattering still remains. This highlights the need to further incorporate macroscopic effects such as the roughness of the snow surfaces into the models (Leroux and Fily, 1998; Hudson and Warren, 2007).

Ground-based instruments observe the directional reflectance of a characteristic, homogenous surface, whereas airborne and

satellite observations average over a larger measurement area. Thus, the latter are more suitable for studying the influence of macroscopic surface roughness on the surface BRDF. Nolin et al. (2002) employed multiangular measurements with the multi-angle imaging spectroradiometer (MISR) for the characterization of surface roughness over Greenland and Antarctica. Measurements with the clouds and the Earth's radiant energy system (CERES) showed monthly regional biases between -12 and 7.5 W m$^{-2}$ in the cloudless TOA solar irradiance. These biases were attributed to the effect of sastrugi, which introduce

a significant solar azimuth dependence: Kuchiki et al. (2011) observed a diurnal cycle in MODIS reflectances over the South Pole. In general, sastrugi decrease the forward scattering by casting shadows, increase the backward scattering due to a lower effective incident angle caused by the sastrugi slope, and the snow BRDF loses its azimuthal symmetry (Warren et al., 1998). Zhuravleva and Kokhanovsky (2011) observed a larger effect for a higher density of the sastrugi field.

Airborne measurements can be employed for the validation of BRDF models and the comparison with the large pixel size of

satellite observations. Gatebe and King (2016) provided an extensive database of airborne spectral BRDFs for various surface types, e.g., ocean, vegetation, snow, desert, and clouds. The BRDFs were acquired by the Cloud Absorption Radiometer (CAR, Gatebe et al., 2005) over a 30-year period between 1984 and 2014. The CAR is a scanning radiometer covering 14 spectral channels between 480 and 2324 nm. The effect of surface roughness on the BRDF was studied by Lyapustin et al. (2010) with CAR measurements during the Arctic Research of the Composition of the Troposphere from Aircraft and Satellites (ARC-

TAS) spring campaign in April 2008 (Jacob et al., 2010; Matsui et al., 2011). Their results showed an agreement between the kernel-based Ross-Thick Li-Sparse-Reciprocal (RTLSR) BRDF model (Lucht et al., 2000) used in the operational MODIS BRDF/albedo product (Schaaf et al., 2002) and the CAR measurements to within ± 0.05. The RTLSR model performed better in the forward scattering direction, whereas the kernel-based Modified Rahman-Pinty-Verstraete (MRPV) BRDF model (Martonchik et al., 1998) used for the processing of data from MISR performed better in the backscattering direction. Jiao

et al. (2019) proposed an additional snow kernel within the kernel-driven BRDF model framework to better account for the



anisotropic reflection of pure snow surfaces. Incorporating the additional snow kernel yielded a correlation coefficient of 0.9 (compared to 0.65 for the original 3-kernel model) and only small biases between the model and different BRDF validation data from ground-based and satellite observations.

Cox and Munk (1954) analyzed radiance calibrated analog photographs for the parameterization of the ocean BRDF. Nowa-
days, digital cameras are increasingly applied in vegetation and soil monitoring (Lebourgeois et al., 2008). The instantaneous measurement of multiple viewing angles facilitates aerial BRDF measurements with digital cameras. The high angular resolution allows the detection of features in the anisotropy of the reflection, which could be missed with HDRF measurements allowing only a limited number of observation angles. This was demonstrated by Goyens et al. (2018) for ground-based measurements of the snow HDRF comparing radiance measurements with a spectroradiometer at discrete viewing angles with
hyperangular observations using a fish-eye radiance camera. Ehrlich et al. (2012) used a commercial digital camera equipped with a wide-angle lens with a FOV of 100° for measurements of the HDRF of snow-covered sea ice, ocean, and clouds (viewing zenith angle up to around 60°).

This study presents a method to derive the snow HDRF at visible wavelength (spectral band 490-585 nm) from airborne digital camera measurements in Antarctica during austral summer in 2013/14. Concurrent measurements of the optical-equivalent
snow grain size retrieved from spectral surface albedo measurements, and surface roughness determined by means of a laser scanner, allow for investigating their effect on the snow HDRF in separate case studies. Subsequently, the measurements are used to parameterize the snow HDRF applying the RTLSR model (Lucht et al., 2000). The presented methodological approach allows for a direct comparison of the airborne HDRF measurements with satellite observations from MODIS.

The definition of reflectance quantities used within this work, the modeling of the BRDF, and the inversion of a semi-
empirical, kernel-driven BRDF model are presented in Sect. 2. The field work and instrumentation are presented in Sect. 3 together with the detailed calibration involved in the measurement of the snow HDRF. The dependence of the snow HDRF with respect to the solar zenith angle, the surface roughness, and the optical-equivalent snow grain size is studied in Sect. 4 based on three cases. Subsequently, the airborne measurements are compared with satellite-derived BRDF from MODIS. In the concluding Sect. 5, the findings of this work are summarized and perspectives for future studies are given.

## 2   Methodology

### 2.1   Definition of reflectance quantities

The definition of the reflectance quantities applied within this work follows Schaepman-Strub et al. (2006). The intrinsic reflectance properties of a surface are given by its BRDF. It quantifies the reflection of the incident radiation at the surface and its scattering from one into another direction of the hemisphere. The spectral BRDF ($f_{\mathrm{BRDF}}$, unit of $\mathrm{sr}^{-1}$) provides for each
solar zenith ($\theta$) and azimuth angle ($\varphi$) of incident direct irradiance $F_{\mathrm{i}}\left(\theta_{\mathrm{i}},\varphi_{\mathrm{i}};\lambda\right)$ the reflected radiance for all reflection angles $I_{\mathrm{r}}\left(\theta_{\mathrm{i}},\varphi_{\mathrm{i}};\theta_{\mathrm{r}},\varphi_{\mathrm{r}};\lambda\right)$ by

$$f_{\mathrm{BRDF}} = \frac{\mathrm{d}I_{\mathrm{r}}\left(\theta_{\mathrm{i}},\varphi_{\mathrm{i}};\theta_{\mathrm{r}},\varphi_{\mathrm{r}};\lambda\right)}{\mathrm{d}F_{\mathrm{i}}\left(\theta_{\mathrm{i}},\varphi_{\mathrm{i}};\lambda\right)}. \tag{1}$$





The bidirectional reflectance factor (BRF, dimensionless) is obtained when the BRDF of a sample surface is divided by the BRDF of a diffuse (Lambertian) standard surface illuminated under the same conditions (identical beam geometry). The BRF is given by

$$f_{\mathrm{BRF}} = \frac{\mathrm{d}I_{\mathrm{r}}\left(\theta_{\mathrm{i}}, \varphi_{\mathrm{i}}; \theta_{\mathrm{r}}, \varphi_{\mathrm{r}}\right)}{\mathrm{d}F_{\mathrm{i}}\left(\theta_{\mathrm{i}}, \varphi_{\mathrm{i}}\right)} \cdot \frac{\mathrm{d}F_{\mathrm{i}}\left(\theta_{\mathrm{i}}, \varphi_{\mathrm{i}}\right)}{\mathrm{d}I_{\mathrm{r}}^{\mathrm{ideal}}\left(\theta_{\mathrm{i}}, \varphi_{\mathrm{i}}\right)} = \pi \cdot f_{\mathrm{BRDF}}. \tag{2}$$

5    In Eq. 2 and in the remainder of this section, the spectral dependence is omitted for reasons of simplicity. The definition of the HDRF is analogous to the BRF, but includes irradiance from the entire hemisphere (denoted with $2\pi$):

$$f_{\mathrm{HDRF}} = \frac{\mathrm{d}I_{\mathrm{r}}\left(\theta_{\mathrm{i}}, \varphi_{\mathrm{i}}, 2\pi; \theta_{\mathrm{r}}, \varphi_{\mathrm{r}}\right)}{\mathrm{d}I_{\mathrm{r}}^{\mathrm{ideal}}\left(\theta_{\mathrm{i}}, \varphi_{\mathrm{i}}, 2\pi\right)}$$

$$= f_{\mathrm{BRF}}(\theta_0, \varphi_0; \theta_{\mathrm{r}}, \varphi_{\mathrm{r}}) \cdot f_{\mathrm{dir}} + f_{\mathrm{BRF}}(2\pi; \theta_{\mathrm{r}}, \varphi_{\mathrm{r}}) \cdot (1 - f_{\mathrm{dir}}). \tag{3}$$

$f_{\mathrm{dir}}$ denotes the fraction of direct incident radiation (i.e. $f_{\mathrm{dir}} \in [0, 1]$). The additional integration over all reflection angles leads to the bihemispherical reflectance, generally called surface albedo $\alpha$.

## 2.2    Modeling of the bidirectional reflectance

The shape of the BRDF is described by a weighted sum of trigonometric functions, generally referred to as kernels for volumetric scattering ($K_{\mathrm{vol}}$), geometric scattering ($K_{\mathrm{geo}}$), and isotropic scattering ($K_{\mathrm{iso}}$).

The kernel-driven semi-empirical Ross-Li model (Lucht et al., 2000), which forms the basis of the MODIS 16-day BRDF/albedo product (Schaaf et al., 2002), is applied within this study. The BRDF is given as a linear combination of the kernels with corresponding non-negative weighting functions $f_{\mathrm{iso}}$, $f_{\mathrm{vol}}$, and $f_{\mathrm{geo}}$:

$$f_{\mathrm{BRDF}}\left(\theta_{\mathrm{r}}, \theta_0, \Delta\varphi, \lambda\right) = f_{\mathrm{iso}}(\lambda) \cdot K_{\mathrm{iso}} + f_{\mathrm{vol}}(\lambda) \cdot K_{\mathrm{vol}}\left(\theta_{\mathrm{r}}, \theta_0, \Delta\varphi\right) + f_{\mathrm{geo}}(\lambda) \cdot K_{\mathrm{geo}}\left(\theta_{\mathrm{r}}, \theta_0, \Delta\varphi\right), \tag{4}$$

with the viewing zenith angle $\theta_{\mathrm{r}}$, solar zenith angle $\theta_0$, relative viewing azimuth angle $\Delta\varphi$, and wavelength $\lambda$. The kernels,

$$K_{\mathrm{iso}} = 1, \tag{5}$$

$$K_{\mathrm{vol}} = \frac{\left(\frac{\pi}{2} - \theta\right)\cos\theta + \sin\theta}{\cos\theta_0 + \cos\theta_{\mathrm{r}}} - \frac{\pi}{4}, \quad \text{and} \tag{6}$$

$$K_{\mathrm{geo}} = \mathcal{O}\left(\theta_{\mathrm{r}}, \theta_0, \Delta\Phi\right) - \sec\theta_0 - \sec\theta_{\mathrm{r}} + \frac{1}{2}\left(1 + \cos\theta\right) \cdot \sec\theta_0 \cdot \sec\theta_r, \tag{7}$$

depend on the scattering angle $\theta$ and the function $\mathcal{O}$:

$$\cos\theta = \cos\theta_0 \cdot \cos\theta_{\mathrm{r}} + \sin\theta_0 \cdot \sin\theta_{\mathrm{r}} \cdot \cos\Delta\Phi, \tag{8}$$

$$\mathcal{O}\left(\theta_{\mathrm{r}}, \theta_0, \Delta\Phi\right) = \frac{1}{\pi}\left(\mathcal{C} - \sin\mathcal{C} \cdot \cos\mathcal{C}\right) \cdot \left(\sec\theta_0 + \sec\theta_{\mathrm{r}}\right). \tag{9}$$

The functions $\mathcal{C}$ and $\mathcal{D}$ depend solely on the viewing and illumination geometry:

$$\cos\mathcal{C} = \frac{2 \cdot \sqrt{\mathcal{D}^2 + \left(\tan\theta_{\mathrm{r}} \cdot \tan\theta_0 \cdot \sin\Delta\Phi\right)^2}}{\sec\theta_0 + \sec\theta_{\mathrm{r}}}, \tag{10}$$

$$\mathcal{D} = \sqrt{\tan^2\theta_{\mathrm{r}} + \tan^2\theta_0 - 2\tan\theta_{\mathrm{r}} \cdot \tan\theta_0 \cdot \cos\Delta\Phi}. \tag{11}$$





Originally, the concept of the model was developed studying typical features in observations of the bidirectional reflectance of various surface types (Roujean et al., 1992). First, bare soil surfaces exhibit strong backscattering characteristics and show the effect of geometrical structures on the surface. Secondly, dense leaf canopies typically feature a minimum reflectance close to the nadir-direction that increases off-nadir for all azimuthal directions. In their approach, Roujean et al. (1992) represented

the BRDF model as a linear combination of these two observational characteristics; the former being accounted for by $K_{\mathrm{geo}}$, the latter by $K_{\mathrm{vol}}$. The volumetric kernel $K_{\mathrm{vol}}$ stems from volume scattering radiative transfer models (Ross, 1981). Thereby, randomly located facets that absorb and scatter incident radiation are modeled under the single-scattering approximation. The formula given in Eq. 6 corresponds to the Ross-Thick kernel for a dense leaf canopy. The geometric kernel $K_{\mathrm{geo}}$ is derived from surface scattering and geometric shadow casting theory (Li and Strahler, 1992) and expresses effects caused by intercrown

gaps within vegetation. It represents randomly oriented vertical protrusions on a flat and horizontal surface that isotropically reflect radiation. Eq. (7) gives the Li-Sparse geometric kernel in reciprocal form describing the casting of shadows by a sparse ensemble of surface objects.

Hu et al. (1997) validated kernel-based BRDF models with 27 multi-angular data sets from various land cover types. The accuracy of the model was high. The correlation coefficient between model and observations was above 0.7 for all and above

0.9 for more than half of the data sets. Although originally calculated for surfaces covered with vegetation, the Ross-Thick and Li-Sparse-Reciprocal kernels are also applied for snow surfaces for the MODIS BRDF/albedo product. Stroeve et al. (2005) and Stroeve et al. (2013) assessed the accuracy of the 16-day albedo product of eleven years of measurements at 17 automatic weather stations on the Greenland ice sheet. They retrieved physically realistic ice sheet albedo values with an overall mean bias between MODIS and the in situ measurements of 0.022.

## 2.3 Inversion of semi-empirical, kernel-driven model

The main benefit of retrieving the weighting functions $f_{\mathrm{iso}}$, $f_{\mathrm{vol}}$, and $f_{\mathrm{geo}}$ from the HDRF measurements is that they encompass all of the angular radiance information. The retrieval is done by inverting the RTLSR BRDF model. Thus, information about the complete two-dimensional shape of the HDRF is used.

The modeled $f_{\mathrm{BRDF}}(\theta_{\mathrm{r}}, \theta_0, \Delta\Phi, \lambda)$ from Eq. 4 can be written in the form of a sum as:

$$f_{\mathrm{BRDF},l} = \sum_{k=1}^{3} f_k \cdot K_{kl}. \tag{12}$$

The spectral dependence is omitted here and the index $l$ denotes a particular viewing and illumination geometry $(\theta_{\mathrm{r}}, \theta_0, \Delta\Phi)_l$. Considering an observation with $N$ directional measurements $\rho_l$ ($l = 1, \ldots, N$), the error function $\mathcal{E}^2$ is defined as the difference between the observed and the modeled reflectances such that,

$$\mathcal{E}^2 = \frac{1}{d} \sum_{l=1}^{N} \frac{(\rho_l - f_{\mathrm{BRDF},l})^2}{w_l}. \tag{13}$$

The degree of freedom $d$ is $(N-3)$ and $w_l$ denotes weights which are assigned to the respective observations. In general, $w_l$ could take the values 1, $\rho_l$, or $\rho_l^2$. The goal of the inversion is the determination of the model weighting functions $f_k$ such





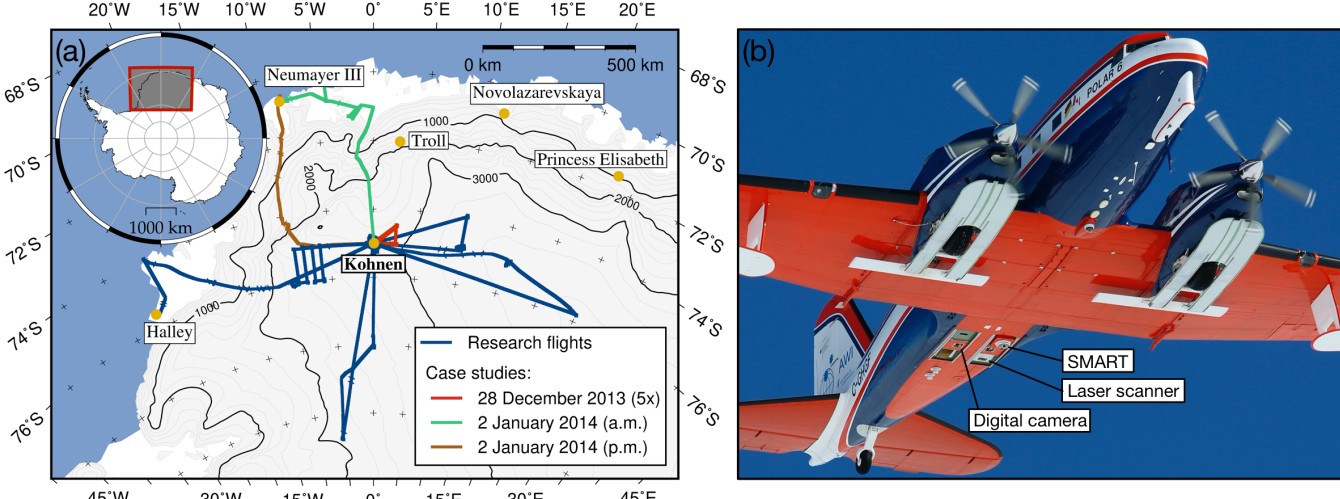

**Figure 1. (a)** Illustration of research flights with Polar 6 in Dronning Maud Land between 24 December 2013 and 5 January 2014. The five flights on 28 December 2013 (marked in red) and the two flights on 2 January 2014 (green and brown) are used for the case studies. The parameter ranges covered by the case studies are marked with the same colors within the histograms in Fig. 2. **(b)** Photograph of Polar 6 illustrating the mounting of the digital camera, SMART, and the laser scanner.

that $\mathcal{E}^2$ is minimized. Strahler et al. (1996), amongst others, presented the analytical solutions for the three $f_k$ following this minimization and Lewis (1995) showed that $\mathcal{E}^2$ has a global minimum in $f_k$.

The inversion depends on the choice of the error function $\mathcal{E}^2$ and, thus, on the choice of the weights $w_l$. Minimizing the absolute error (using weights equal to unity) leads to smaller errors in angular domains with a large reflectance. In contrast, minimizing the relative error (e.g., $w_l = \rho_l, \rho_l^2$) performs better in angular regions with lower reflectance. Subsequently, different choices for $w_l$ were tested for the retrievals performed in this work and $w_l = \rho_l^2$ was chosen as it produced the lowest retrieval errors across the entire angular domain.

## 3 Measurements and instrumentation

The airborne measurements were performed with the Polar 6 research aircraft (Wesche et al., 2016) operated by the Alfred Wegener Institute, Helmholtz Centre for Polar and Marine Research (AWI), Bremerhaven, Germany. Polar 6 was based at the Kohnen research station ($75°0'$ S, $0°4'$ E). Figure 1a illustrates the tracks of the research flights with Polar 6 (60 flight hours) covering an area of $1000 \times 1000 \, \mathrm{km}^2$ around Kohnen station in Dronning Maud Land. The flights were performed between 24 December 2013 and 5 January 2014. The observations comprised regions with higher (coastal areas) and lower precipitation amounts and frequencies (inner Antarctica), as well as a variety of surface roughness structures. An overview of the airborne instrumentation is given in Carlsen et al. (2017); the mounting of the digital camera, laser scanner, and the Spectral Modular Airborne Radiation measurement sysTem (SMART) for surface albedo measurements on the underside of the aircraft





is illustrated in Fig. 1b. The variability of the solar zenith angle is illustrated in Fig. 2a in the form of a histogram. During the flights, the solar zenith angle varied between 49° and 73°.

## 3.1 Surface roughness measurements using a laser scanner

The snow surface topography was measured using the airborne laser scanner RIEGL VQ580. In time-of-flight laser ranging, a

near-infrared laser beam (1064 nm wavelength) is emitted downward and subsequently reflected upward by the snow surface before the echo is acquired by the sensor. From the time lag between emission and detection, the distance to the ground is calculated with a precision of about 25 mm depending on flight altitude. A fast-rotating polygonal mirror with a FOV of 60° and 10 to 150 scans per second allows for fully linear, unidirectional, and parallel scan lines.

After a correction for the aircraft attitude, a $1\times1\,\mathrm{km}^2$ digital elevation model (DEM) is generated from the geotagged laser

point cloud with a resolution of 1 m. Subtracting the large scale topography (smoothed DEM), the residual field contains the roughness information. The standard deviation of the residual field is interpreted as the surface roughness at the central coordinate of the DEM. Thus, roughness data is given for one data point per 1 km along the flight track of the aircraft. The uncertainty of the absolute height measurements (used for DEM generation) is less than 0.1 m. The relative analysis applied to the measurements yields an even higher accuracy.

The range in surface roughness is illustrated in Fig. 2b in the form of a histogram. The snow surface was generally smooth with roughness structures mostly below 5-10 cm (note the logarithmic scale, maximum: 2.2 m).

## 3.2 Optical-equivalent snow grain size as retrieved from spectral surface albedo measurements

Solar spectral radiation measurements were conducted using SMART (Wendisch et al., 2001; Ehrlich et al., 2008). The upward and downward spectral irradiance $[F^{\uparrow}(\lambda), F^{\downarrow}(\lambda)]$ is measured within 0.3 to 2.2 μm wavelength applying an active horizontal

adjustment system of the optical inlets to compensate for aircraft movement. From the irradiance measurements, the spectral snow surface albedo was obtained (Wendisch et al., 2004). The spectral resolution is 2 to 3 nm between 0.3 and 1.0 μm and 15 nm up to 2.2 μm wavelength; the temporal resolution is in the order of 1 s. Combining the errors associated with the signal-to-noise ratio (1.3-3.0 %), the accuracy of the dark correction (0.1 %), the wavelength calibration (1.0 %), the accuracy of the cross-calibration (1.0-4.5 %), the non-ideal cosine response of the optical inlets (3.5 %), and the horizontal stabilization

(1.0 %), the surface albedo measurements with SMART have an estimated uncertainty between 4.1 % and 8.1 % depending on wavelength (Carlsen et al., 2017).

The optical-equivalent snow grain size is defined as the radius $R_{\mathrm{opt}}$ of a collection of spheres with the same volume-to-surface ratio compared to the actual non-spherical snow grains. The retrieval of $R_{\mathrm{opt}}$ from spectral surface albedo measurements is described in Carlsen et al. (2017). In principle, the snow grain size and pollution amount (SGSP) algorithm (Zege

et al., 2011) was extended to spectral ratios of surface albedo at 1280 and 1100 nm wavelength. Being independent of systematic measurement uncertainties (e.g., cross-calibration of the optical inlets), this approach decreases the uncertainty of the retrieved $R_{\mathrm{opt}}$ compared to the single-wavelength approach. The retrieved $R_{\mathrm{opt}}$ from SMART and analogous ground-based measurements were validated against grain size observations utilizing reflectance measurements with MODIS (Carlsen et al.,

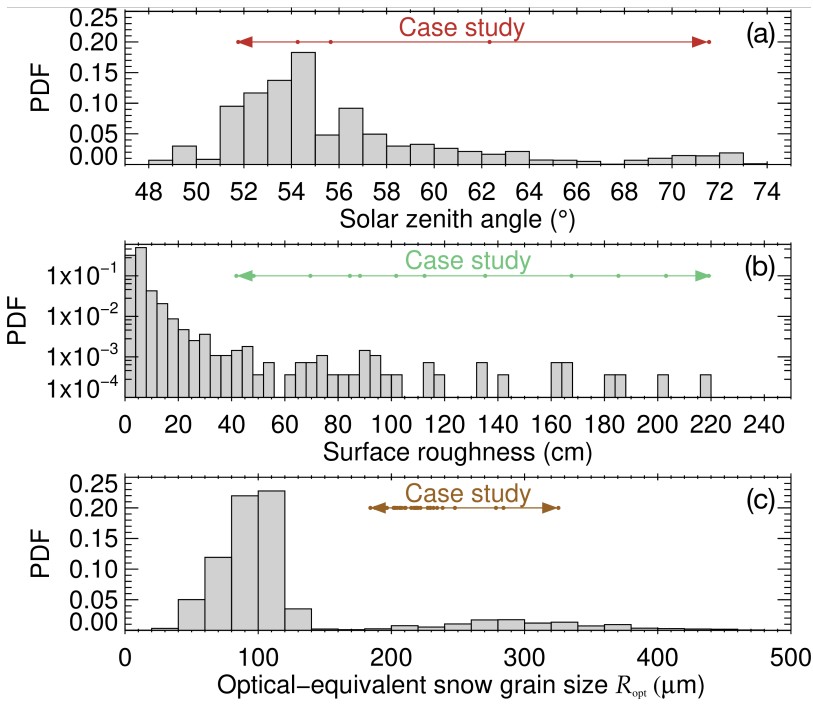

**Figure 2.** Variability of different parameters during research flights shown as histograms of the probability density function (PDF). The parameter ranges covered by the case studies are marked with the same colors as the corresponding flight tracks in Fig. 1. **(a)** Variability of solar zenith angle (binsize: 1°). **(b)** Variability of surface roughness (binsize: 4 cm). Note the logarithmic scale. **(c)** Variability of optical-equivalent snow grain size (binsize: 20 μm).

2017). The grain size as retrieved from SMART measurements (mean value: 105 μm) is slightly higher than the MODIS retrievals (89 μm). The larger dataset of ground-based observations (72 μm) agrees well with the MODIS retrievals within the ranges given by the measurement uncertainties (linear correlation coefficient: 0.78, root-mean-square error: 24 μm).

The variability of $R_{\mathrm{opt}}$ during the research flights is illustrated in Fig. 2c in the form of a histogram. The optical-equivalent
5 snow grain size varied between 16 and 480 μm, but was mostly below 120 μm depending on precipitation and snow metamorphism processes.

### 3.3 Directional radiance measurements using a digital camera

#### 3.3.1 Camera specifications

The digital camera Canon EOS-1D Mark III was used for the airborne HDRF measurements of the snow surface. It is a digital
10 single-lens reflex camera with a complementary metal oxide semiconductor (CMOS) image sensor. The CMOS sensor covers
$3908 \times 2600$ picture elements (pixels) on a sensor area of $28.1 \times 18.7$ mm (Advanced Photo System APS-H format). During the observations, the camera was configured with the 8 mm F3.5 EX DG Circular Fish-eye lens by Sigma. Color information





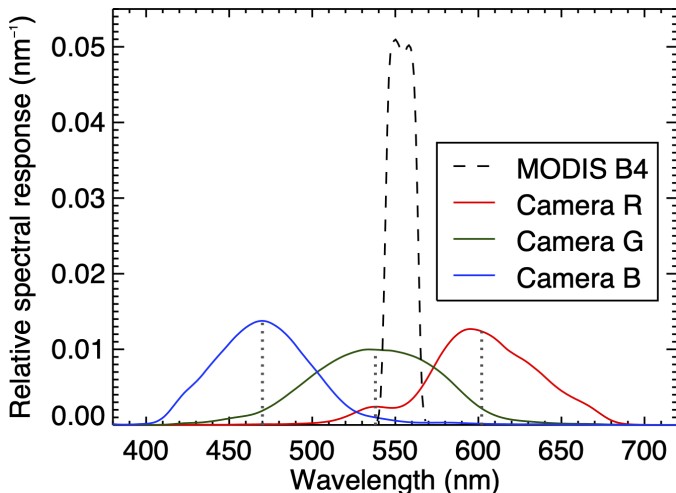

**Figure 3.** Relative spectral response functions for the three camera channels (colored lines: R, G, and B). The vertical dotted lines denote the respective central wavelengths. Black dashed line: RSR function for MODIS spectral band 4.

is obtained by means of a color filter array (CFA) in front of the CMOS sensor. The CFA consists of red (R), green (G), and blue (B) color filters, which determine the spectral response of the underlying pixel.

In principle, the measured signal $s$ (in digital numbers, DN) should be saved essentially verbatim in the proprietary Raw format CR2 (Canon Raw version 2). This would ensure highest flexibility in the data analysis as no interpolation or white balance color correction is applied to the raw data. Full control over the post-processing steps is a prerequisite for radiometric measurements. During the airborne measurements, the camera was set to produce sRaw output format (small raw) with a resolution of $1944 \times 1296$ pixels. At some point, internal chrominance subsampling is applied by the camera (Kerr, 2015).

The sRaw photos are decoded utilizing the open source tool *dcraw* (Coffin, 2017). The darkness level was set to 0 DN (see below) and the saturation level to 16383 DN as the camera captures images with color depth of 14 bit. In between, the raw data is linearly interpolated. The multipliers for all channels are set to one meaning no white balance color correction is applied. The dynamic range of the output file is 16 bit (saturation at 65536 DN).

To characterize the linearity of the sensor, photos from the radiation emitted by an integrating sphere with varying output intensities were taken with the camera in the laboratory (Carlsen, 2018). The camera sensor showed a linear sensitivity over a large part of the dynamic range up to 53000 DN, the coefficient of correlation was exceptionally high with 0.99998. Photographs that measured higher signals (0.4 % of total number) were excluded from the analysis.

Compared to the dynamic range, the dark current (5 DN) and the read-out noise (7 DN) can be neglected.

Information on the spectral response of the camera is needed for the radiometric calibration. It determines the extent to which radiation of a certain wavelength passes the fish-eye lens as well as the CFA and is registered by the photodiodes. In this





regard, the relative spectral response (RSR) function in units $\mathrm{nm}^{-1}$ of the three camera channels is defined as

$$\mathrm{RSR}(\lambda) = \frac{T_\mathrm{c}(\lambda)}{\int\limits_0^\infty T_\mathrm{c}(\lambda)\,\mathrm{d}\lambda}. \tag{14}$$

$T_\mathrm{c}(\lambda)$ with $\mathrm{c = R, G, B}$ denotes the dimensionless spectral transmission coefficients. The RSR function is normalized such that

$$5 \quad \int\limits_0^\infty \mathrm{RSR}(\lambda)\,\mathrm{d}\lambda = 1. \tag{15}$$

The RSR function of the camera was measured in the laboratory using an integrating sphere as a radiation source and varying the outgoing radiation between 300 and 750 nm in increments of 5 nm wavelength by means of a grating monochromator. Figure 3 shows the measured RSR functions for the three camera channels. The central wavelengths of the non-Gaussian RSR functions are 602 nm (R), 538 nm (G), and 470 nm (B). Their full width at half maximum (FWHM) varies between 68 nm for 10   the blue channel and 95 nm for the green channel.

The FOV of the camera is 180° due to the optics of the fish-eye lens. However, the camera is installed slightly above the lower aircraft body so that the camera is protected especially during take-off and landing. Therefore, parts of the aircraft frame are constantly in the FOV of the camera, which is why the effective FOV is reduced to approximately 160°.

### 3.3.2   Radiance calibration and image post-processing

15   The post-processing of each raw image involves (a) the radiometric calibration, (b) the geometric calibration, (c) the aircraft attitude correction, and (d) the calculation of the HDRF. A detailed description of the different steps can be found in Carlsen (2018).

The measured signal was converted into the physical quantity of radiance (unit of $\mathrm{W\,m^{-2}\,nm^{-1}\,sr^{-1}}$) by means of a radiometric calibration in the laboratory. The pixel respond differently to an uniform illumination due to manufacturing tolerances 20   (e.g., irregularities in the used silicon), contamination with dust particles, and optical effects at the edges of the lenses. These deviations lead to the photo response non-uniformity (PRNU) and need to be corrected.

The camera was positioned in front of an integrating sphere that served as an uniform radiation source. However, a distinct decrease in brightness is visible from the center to the edges of the sensor. This vignetting effect is typical for digital cameras (e.g., Lebourgeois et al., 2008). From the data sheet, the spectral radiance $I_\mathrm{sphere}(\lambda)$ that the integrating sphere emits at the 25   specific optometer current is known. The calibration factor $k_\mathrm{c}$ is defined at each pixel location $(x, y)$ as

$$k_\mathrm{c}(x,y) = \frac{I_\mathrm{sphere}(\lambda)}{s(x,y)} \cdot t_\mathrm{exp}, \tag{16}$$

and carries the unit of $\mathrm{W\,m^{-2}\,nm^{-1}\,sr^{-1}\,(DN/s)^{-1}}$. During the calibration, the exposure time $t_\mathrm{exp}$ was set to 1/1000 s. Not only does $k_\mathrm{c}$ correct for the PRNU, it simultaneously performs the absolute calibration transforming the measured digital signal into the physical quantity of radiance with units $\mathrm{W\,m^{-2}\,nm^{-1}\,sr^{-1}}$.



The geometric calibration of the camera-lens system relates each sensor pixel to a viewing zenith and azimuth angle ($\theta_v$, $\varphi_v$). Often, the process of geometric calibration of a camera involves calibration equipment or the use of planar targets such as checkerboard patterns (e.g., Tsai, 1987; Urquhart et al., 2016). Within this work, a stellar calibration method is applied (e.g., Schmid, 1974; Klaus et al., 2004; Mori et al., 2013; Urquhart et al., 2016) utilizing the high precision to which the positions

of stellar objects are known. In this regard, 684 different star positions identified in subsequent pictures of the night sky were utilized to calculate $\theta_v$ and $\varphi_v$ for each sensor pixel.

As the camera is fixed to the aircraft frame, a correction for the aircraft attitude was implemented. Thus, beside the geometric calibration, the viewing angles $\theta_v$ and $\varphi_v$ of the camera are determined by the attitude angles of the aircraft. The movement of the aircraft needs to be corrected to obtain the reflection angles $\theta_r$ and $\varphi_r$. In this regard, Euler rotations were applied as

described in Ehrlich et al. (2012).

### 3.4 Calculation and uncertainties of measured surface HDRF

The camera took pictures with a temporal resolution of about 8 s. Exemplarily, Fig. 4 demonstrates the derivation of the snow HDRF from a raw image taken by the camera on 2 January 2014 (8:16 UTC, see Fig. 4a). First, the pixels receiving radiation from the direction of the aircraft frame are excluded. For each pixel location ($x$, $y$), the radiance $I(x,y)$ (in units

$\mathrm{W\,m^{-2}\,nm^{-1}\,sr^{-1}}$) is calculated from the measured signal $s$ using the absolute calibration factor $k_c$ and the exposure time $t_{exp}$:

$$I(x,y) = \frac{s(x,y)}{t_{exp}} \cdot k_c(x,y) \tag{17}$$

The camera viewing angles are calculated from the geometric calibration for each pixel. Utilizing the data from the internal navigation system (INS) and the global positioning system (GPS) on Polar 6, the viewing angles are corrected depending on

the roll, pitch and yaw angles of the aircraft at the time of measurement. Finally, the HDRF is calculated by:

$$f_{HDRF}(\theta_r, \varphi_r) = \frac{\pi \cdot I(\theta_r, \varphi_r)}{F^\downarrow} \,. \tag{18}$$

Based on the resulting reflection angles, a polar plot of the measured $f_{HDRF}$ is created (camera channel G, see Fig. 4b). To achieve comparability, each image is rotated into the azimuthal direction of the Sun.

Note that the downward irradiance measurements from SMART could not be used for the calculation of the HDRF due

to calibration issues. Instead, the global irradiance was simulated along the flight track with the library for radiative transfer libRadtran (Mayer and Kylling, 2005) using the discrete ordinate radiative transfer solver DISORT by Stamnes et al. (1988). Vertical profile data of air temperature and relative humidity measured by radiosondes at Kohnen, Neumayer III (König-Langlo, 2014), and Halley (Durre et al., 2016) research stations were used as input for the simulations. The simulated irradiance was integrated over the wavelength range of each camera channel and weighted with the RSR function of the camera. The use of

simulations limits the validity of absolute values of the measured HDRF to cloudless conditions. However, within this work mainly the shape of the HDRF is analyzed which is independent from the absolute value of $F^\downarrow$.





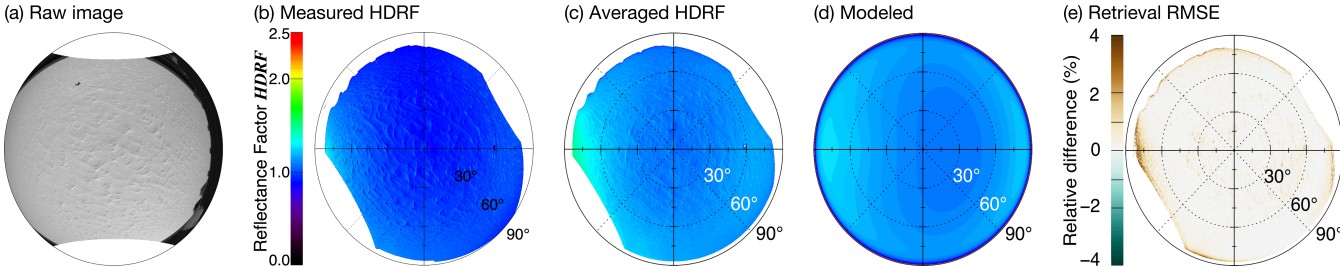

**Figure 4.** Exemplary post-processing and retrieval of model parameters for HDRF measurement. **(a)** Raw image taken on 2 January 2014 at 8:16 UTC ($\theta_0 = 58.9°$). **(b)** Polar plot of observed snow HDRF $\rho_l$ calculated from raw image for camera channel G. The image is rotated into the azimuthal direction of the Sun such that the Sun is on the left. **(c)** Averaged HDRF measurement using 3 images, the footprint is 1670 m. **(d)** Modeled snow HDRF $f_{\mathrm{HDRF},l}$ after retrieval of model parameters $f_{\mathrm{iso}} = 1.12$, $f_{\mathrm{vol}} = 0.17$, and $f_{\mathrm{geo}} = 0.01$ from (c). **(e)** Relative difference $(\rho_l - f_{\mathrm{HDRF},l})/f_{\mathrm{HDRF},l}$. The accumulated RMSE is 0.04.

The overall relative uncertainty of the HDRF measurements with the digital camera range in the order of 4.5 %. The error in the absolute value of $f_{\mathrm{HDRF}}$ might be higher depending on the atmospheric conditions due to the usage of simulated values for the global irradiance $F^\downarrow$ in Eq. (18). The uncertainties in the HDRF measurements stem from sensor characteristics (estimated with 0.5 % due to signal-to-noise ratio, dark current, linearity, read-out noise, chrominance subsampling), the radiometric

calibration (4 %), the geometric calibration and the correction for the aircraft attitude (combined estimate of 1.0 %).

## 3.5    Averaging

From trigonometric considerations, the footprint of the camera (twice the radius $r$ of the disc on the ground pictured by the camera) depends on the flight altitude $z$ and the FOV. At an altitude of 100 m, the radius is 570 m and the footprint approximately 1 km. The footprint grows with increasing altitude to 5.7 km ($z = 500$ m) and 11.3 km ($z = 1000$ m). To get

independent of local roughness features, averaging is performed over time intervals of around 30 s (including 3 to 4 pictures). This interval was chosen trading off the gained independence from local features against a growing footprint and thus reduced comparability with the MODIS BRDF measurements that are provided on a 500×500 m grid. Generally, averaging over 30 s yields footprints between 1 and 2 km. However, individual footprints depend on flight altitude, exact averaging time, and aircraft velocity. The averaged HDRF for the exemplary measurement on 2 January 2014 for a footprint of 1670 m is shown in

Fig. 4c.

## 3.6    Quality of the inversion

An exemplary inversion is performed on the snow HDRF measurement from 2 January 2014 at 8:16 UTC ($\theta_0 = 58.9°$). Figure 4c shows the averaged observations that are used as input for the algorithm. The inversion leads to the retrieved model parameters of 1.12 ($f_{\mathrm{iso}}$), 0.17 ($f_{\mathrm{vol}}$), and 0.01 ($f_{\mathrm{geo}}$). With the calculated kernels $K_{kl}$, the modeled HDRF is obtained (see

Fig. 4d). The shapes of modeled and measured HDRF are similar; their relative difference is mostly within the range of $\pm 5\%$





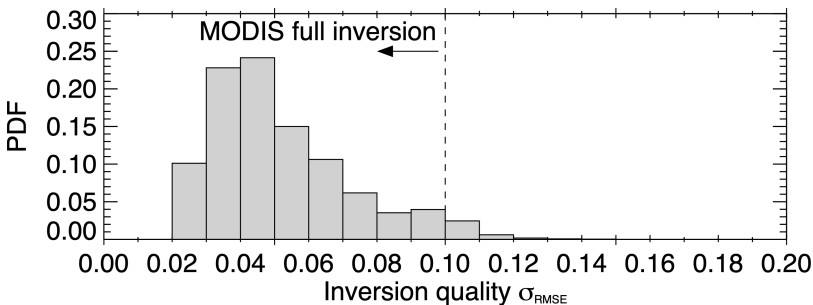

**Figure 5.** Statistics of the quality of the inversion. RMSE of all airborne retrievals (PDF, binsize: 0.01). Vertical dashed line shows RMSE threshold of 0.1 for MODIS full inversions.

(see Fig. 4e). The largest deviations occur in the forward scattering direction (up to 3 %) because the location of maximum is not mimicked perfectly by the modeled HDRF. However, up to viewing zenith angles of around 75°, the relative difference is below 1 %.

The root-mean-square error (RMSE),

$$\sigma_{\mathrm{RMSE}} = \mathcal{E} = \sqrt{\frac{1}{N-3} \cdot \sum_{l=1}^{N} \frac{(\rho_l - f_{\mathrm{HDRF},l})^2}{w_l}}, \tag{19}$$

serves as a criterion for the quality of the inversion. Only RMSE below 0.1 are considered full inversions within the MODIS retrieval (e.g., Stroeve et al., 2005). For the example shown in Fig. 4, it is 4.0 %. $\sigma_{\mathrm{RMSE}}$ strongly depends on the anisotropy of the observed HDRF. Figure 5 shows the RMSE for all airborne retrievals as a histogram, with 96 % of the retrievals showing an RMSE below 0.1. Hence, the upper bound for the uncertainty of the inversion is in the range of 10 %.

The influence of the angular sampling on the retrieval of the model parameters can be investigated using the weights of determination (WoD, Lucht and Lewis, 2000), which depend both on the angular sampling and the number of directional measurements $N$. WoDs (also: noise amplification factors) below one indicate a retrieval uncertainty that is smaller than the RMSE of the inversions. Due to the large $N$ and consistent sampling of the viewing hemisphere, the WoD for the model parameters $f_k$ within this study are mostly below 0.0002. For comparison, the sparse angular sampling of the reflectance

measurements from MODIS observations can lead to WoD above one and only model fits with WoD below 2.5 are considered full inversions.

To study the stability of the inversion as a result of the uncertainties in the reflectance measurements, the stability of the retrieval of $f_k$ was studied for a 5 min flight leg over a homogeneous snow surface on 28 December 2013: the surface roughness varied between 2 and 4 cm, $R_{\mathrm{opt}}$ varied between 80 and 90 µm, and the solar zenith angle was around 55.3°. The inversion

was performed for each individual camera picture (no averaging). The mean and standard deviation of the retrieved model parameters along the flight leg were $1.104 \pm 0.030$ ($f_{\mathrm{iso}}$), $0.296 \pm 0.027$ ($f_{\mathrm{vol}}$), and $0.031 \pm 0.003$ ($f_{\mathrm{geo}}$). The same area was sampled five times at different solar zenith angles on 28 December 2013, the influence of $\theta_0$ on the standard deviations of the





$f_k$ is negligible. The mean standard deviations of 0.031 ($f_{\mathrm{iso}}$), 0.020 ($f_{\mathrm{vol}}$), and 0.003 ($f_{\mathrm{geo}}$) are used as error bars for the $f_k$ within this study.

### 3.7 Approximation of surface BRF with HDRF measurements

Note that this inversion yields the parameters of a BRDF model derived from a HDRF measurement as the BRDF is not measurable under atmospheric conditions. However, the atmospheric influence is assumed to be small due to the high surface elevation and the dry, aerosol-free atmosphere on the Antarctic plateau: the mean AOD at Kohnen station was 0.015 between 2001 and 2006 (at 500 nm, Tomasi et al., 2007) and the mean integrated atmospheric water vapor was $1.1 \, \mathrm{kg \, m^{-2}}$ between December 2013 and January 2014 (Carlsen et al., 2017). Simulations with the library for radiative transfer libRadtran (Mayer and Kylling, 2005) of the mean direct fraction of the global irradiance $f_{\mathrm{dir}}$ at Kohnen station between 10 December 2013 and 31 January 2014 indicate rather weak atmospheric scattering effects. For the central wavelengths of the three camera channels, the mean direct fraction is 87 % (for 602 nm wavelength, channel R), 81 % (538 nm, G), and 69 % (470 nm, B). The values for $f_{\mathrm{dir}}$ are even higher when contributions from a small scattering cone around the incident direction are considered. Schaepman-Strub et al. (2006) simulated the difference between HDRF and BRF for snow surfaces for different fractions $f_{\mathrm{dir}}$ at 550 nm wavelength. They found that with an increasing diffuse fraction of the incident irradiance, the shape of the HDRF is smoothed in comparison to the BRF ($f_{\mathrm{dir}} = 1$, see Eq. 3). For $f_{\mathrm{dir}} = 0.8$ at $\theta_0 = 30°$, the shape of the HDRF is still close to that of the BRF. Hence, the measured HDRF shapes can serve as approximations for the intrinsic BRF of the underlying snow surface for the measurements analyzed within this study ($f_{\mathrm{dir}} > 0.8$).

### 3.8 Comparison with BRDF derived from MODIS satellite observations

The use of the RTLSR model in the retrieval of the model parameters $f_k$ from airborne observations provides the framework to compare with satellite observations. The MODIS BRDF/Albedo Model Parameters product (MCD43A1) provides the $f_k$ in 500 m resolution utilizing multi-date, atmospherically corrected, and cloud-cleared input data over a period of 16 days (Schaaf et al., 2002). Version 6 of the MCD43A1 product uses both Terra and Aqua data and is temporally weighted to the 9th day of a 16-day retrieval window, thereby putting higher emphasis on the actual day of interest than previous versions. The combination of MODIS on Terra and Aqua provides a higher number of high quality reflectance measurements, resulting in better temporal and angular sampling. As the retrieval uncertainty increases with larger solar zenith angle, a thorough quality assessment of the satellite data is needed especially in polar regions. Stroeve et al. (2005) compared the MODIS albedo product with in situ observations on the Greenland ice sheet and recommended to only use full inversion data (RMSE < 0.1, WoD < 2.5). Hence, this study restricts the satellite data to full inversions based on the quality flag information provided in the MCD43A2 product (quality flag values 0 or 1). Lower quality magnitude inversions would facilitate archetypal BRDF parameters to improve the retrieval in case of less than 7 high quality reflectance observations or poor angular sampling. The BRDF data from MODIS spectral band 4 (0.545-0.565 μm) is used as it coincides best with the green camera channel (see Fig. 3).

The MODIS data was resampled on the flight track using nearest neighbour with respect to the great circle distance. For the comparison, the aircraft measurements are filtered analogously: only clear-sky observations with RMSE < 0.1 (96 % of



**Figure 6. Top:** Averaged snow HDRF (490-585 nm wavelength) over 30 s-segment during easternmost flight leg for the five consecutive research flights on 28 December 2013. **Below:** Dependence of the retrieved $f_k$ from the solar zenith angle $\theta_0$ for the five flights. The dashed lines represent linear regression fits of the parameters with respect to $\theta_0$. **(a)** The isotropic weight $f_{\mathrm{iso}}$. **(b)** The volumetric weight $f_{\mathrm{vol}}$. **(c)** The geometric weight $f_{\mathrm{geo}}$.

total measurements), WoD $< 2.5$ (valid for all observations), and a solar zenith angle lower than 70° were used. Stroeve et al. (2005) restricted the analysis to $\theta_0 < 75°$, however, the number of comparable observations between MODIS and the aircraft is not reduced by the stricter condition. In fact, only 434 (21 %) MODIS observations can be compared to the 2078 airborne





observations that in principle fulfill the quality requirements. For the remaining observations, no full inversion retrieval could be performed from the MODIS observations.

## 4  Results and discussion

To study the influence of the solar zenith angle, the surface roughness, and the optical-equivalent snow grain size on the snow

HDRF, the effects of the individual parameters need to be separated. For this purpose, several case studies were selected: (a) measurements at different $\theta_0$ (marked with red circles in Fig. 2a, five flights on 28 December 2013), (b) measurements at different surface roughness (marked with green circles in Fig. 2b, morning flight on 2 January 2014), and (c) measurements at different $R_{\mathrm{opt}}$ (marked with brown circles in Fig. 2c, afternoon flight on 2 January 2014). The remaining two parameters were kept constant throughout the individual cases. After discussing the individual cases, the airborne observations are compared

with the BRDF derived from MODIS.

### 4.1  Influence of solar zenith angle

The research flights with Polar 6 on 28 December 2013 were exploited to study the influence of $\theta_0$ on the snow HDRF. Five consecutive flights were conducted in an area northeast of Kohnen (see red flight track in Fig. 1a).

For each flight, the snow HDRF was averaged over a 30 s-segment during the easternmost flight leg; the results are shown in

the top panel of Fig. 6. The solar zenith angle varied between 51.8° and 71.6°. As Polar 6 took identical routes throughout the flight, the respective flight leg always covered the same area. The central points of the five consecutive segments are separated by less than 1 km. Therefore, the surface roughness (retrieved from the laser scanner) of approximately 6 to 7 cm remained constant. Within the retrieval uncertainties, the same holds true for the optical-equivalent snow grain size. $R_{\mathrm{opt}}$ varied between 70 µm and 85 µm. This minimizes a possible influence of $R_{\mathrm{opt}}$ or $l_{\mathrm{rough}}$ on the snow HDRF that would be superimposed on

the effect of $\theta_0$. Cloudless conditions prevailed during the flights.

With increasing $\theta_0$, the maximum of the HDRF in the forward scattering direction becomes more pronounced (see Fig. 6). Correspondingly, the anisotropy gets larger. Figure 6 shows the dependence of the $f_k$ from $\theta_0$ for the green camera channel (490-585 nm wavelength). $f_{\mathrm{iso}}$ shows no clear dependence on $\theta_0$ and varies between 1.06 and 1.10. $f_{\mathrm{geo}}$ weakly increases with $\theta_0$ from 0.03 at $\theta_0 = 51.8°$ to 0.05 at 71.6°. During the first three research flights, $\theta_0$ did vary only slightly. Therefore, the

strongest trends are visible between flights 3 and 5. In particular, $f_{\mathrm{vol}}$ increases from 0.25 to 0.36.

The probability for photons entering the snowpack to leave it after just a few scattering events increases for lower Sun elevation. With increasing $\theta_0$, the reflection properties of the snow layer converge to the single-scattering properties of ice crystals. In addition, longer shadows are cast by surface roughness structures at larger $\theta_0$. This expected increase in the anisotropy of the snow HDRF for increasing $\theta_0$ is obvious in the changing model parameters $f_k$.



**Figure 7. Top:** Two examples of snow HDRF (490-585 nm wavelength) at different times during an overflight over a sastrugi field on 2 January 2014 (morning flight). **Below:** Dependence of the retrieved $f_k$ from the surface roughness. The dashed lines represent linear regression fits of the parameters with respect to $l_{\text{rough}}$. **(a)** The isotropic weight $f_{\text{iso}}$. **(b)** The volumetric weight $f_{\text{vol}}$. **(c)** The geometric weight $f_{\text{geo}}$.

## 4.2 Influence of surface roughness

On 2 January 2014, a research flight was conducted from Kohnen in the direction of the coastline to Neumayer III research station. The flight track is shown in Fig. 1a (green). Between 8:50 and 9:00 UTC, Polar 6 crossed a sastrugi field at an elevation between 100 and 250 m above sea level. This fostered large variations in the surface roughness between 42 cm and 2.2 m which





became distinguishable by visual observation from the aircraft. At the same time, the solar zenith angle was constant at around 55.5°. For these values, only little variation of the snow HDRF with $\theta_0$ was found (compare Fig. 6). The optical-equivalent snow grain size varied between 240 and 320 μm, the flight altitude remained at around 600 m above the surface leading to a footprint of approximately 3.5 to 4.0 km. During this flight leg, 14 measurements were used to study the influence of the

surface roughness on the snow HDRF. Two examples of the snow HDRF are shown in the top panel of Fig. 7.

For the smooth surfaces, the maximum in forward scattering direction is more pronounced. With increasing surface roughness, the anisotropy gets lower. This is in accordance with Warren et al. (1998) who observed a reduction of the forward peak due to sastrugi during tower measurements at South Pole Station. Roughness structures enhance the backscatter by changing the effective angle of incidence. In addition, they cast shadows that reduce the forward scatter. The evolution of the retrieved

model parameters $f_k$ as illustrated in Fig. 7 demonstrates the decreasing anisotropy: $f_{\mathrm{iso}}$ and $f_{\mathrm{geo}}$ show a slight decreasing trend whereas, most prominently, $f_{\mathrm{vol}}$ decreases from 0.33 to 0.24 with increasing surface roughness.

### 4.3   Influence of optical-equivalent snow grain size

The separation of the two parameters $\theta_0$ and $l_{\mathrm{rough}}$ showed that they influence the snow HDRF in opposite ways. Ice absorption becomes dominant in the near-infrared part of the solar EM spectrum. Hence, even though a wide variety of optical-equivalent

snow grain sizes was covered during the flights (16 to 480 μm), no large effect on the snow HDRF was expected as the camera is only sensitive in the visible wavelength range (Warren et al., 1998, compare Fig. 3). However, 32 observations between 14:45 and 15:15 UTC on 2 January 2014 (see brown flight track in Fig. 1a) and depicted in Fig. 8 show a decrease of $f_{\mathrm{vol}}$ from 0.34 at 184 μm to 0.17 at 325 μm snow grain size. This is consistent with the decrease in anisotropy as visible in the averaged HDRF measurements (top panel in Fig. 8).

However, this stands in contrast to findings by Steffen (1987), who measured the bidirectional reflectance on a glacier in Northwestern China for powder snow (grain size 0.15 mm), new snow (0.5-1.0 mm), and old snow (1-3 mm) and found higher anisotropy for larger snow grains. It should be noted that the range of snow grain sizes covered within this study is significantly smaller and the footprint of the measurements is larger compared to the local study by Steffen (1987). Even though other parameters were kept constant during this case study ($\theta_0$: 51.3-53.4°, $l_{\mathrm{rough}}$: 4-7 cm, footprint: 1.3-1.7 km), effects of other

influencing parameters cannot be ruled out completely. For example, Steffen (1987) reported changing ice crystal shapes between the measurements and suggested a possible influence of dust particles within the snow. This could partly explain the opposite conclusions drawn within this study.



**Figure 8. Top:** Two examples of snow HDRF (490-585 nm wavelength) at different times on 2 January 2014 (afternoon flight). **Below:** Dependence of the retrieved $f_k$ from the optical-equivalent snow grain size. The dashed lines represent linear regression fits of the parameters with respect to $R_{opt}$. **(a)** The isotropic weight $f_{iso}$. **(b)** The volumetric weight $f_{vol}$. **(c)** The geometric weight $f_{geo}$.





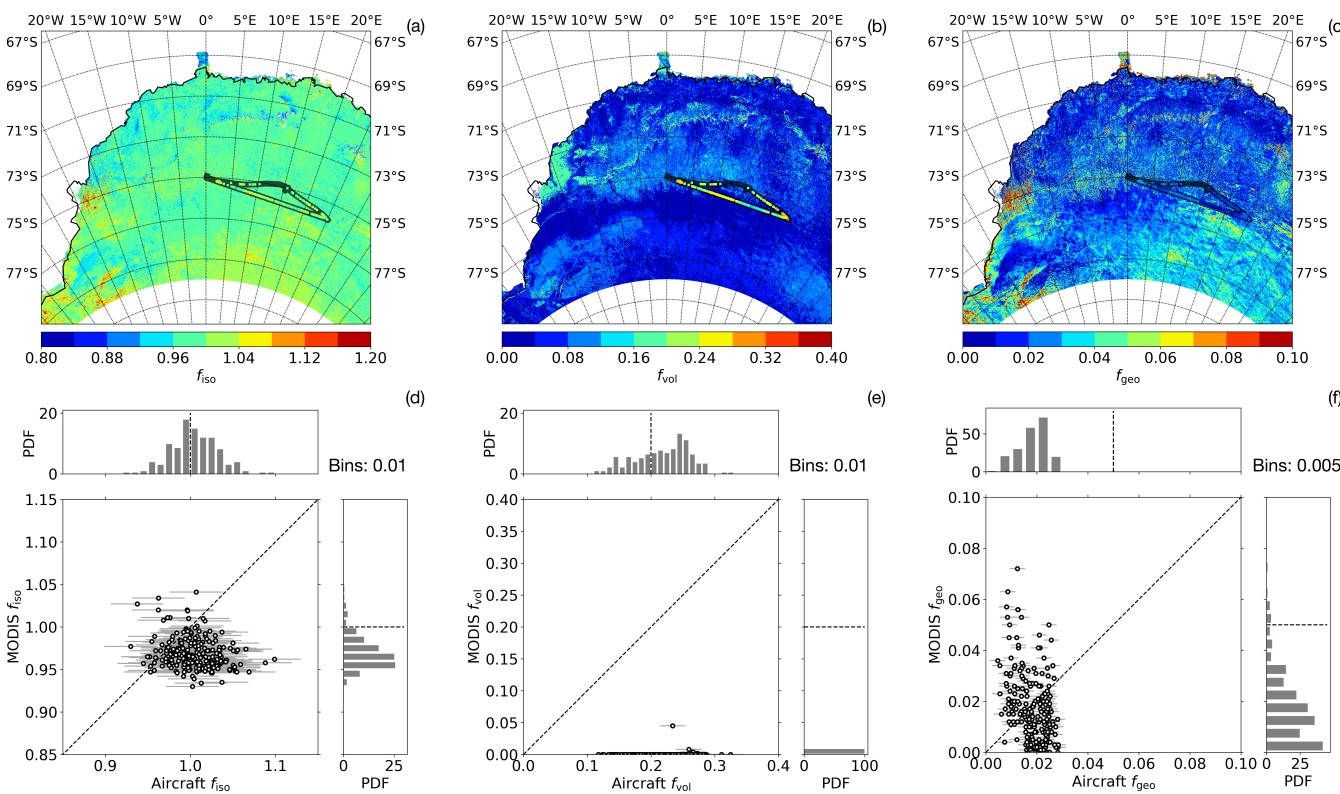

**Figure 9.** Comparison between airborne and MODIS retrievals of $f_k$ (490-585 nm wavelength). **Top: (a)** Map of the area covered by the research flights, the colored contours correspond to the MODIS retrieval of $f_{\mathrm{iso}}$. Black: flight track on 29 December 2013. Colored circles on black flight track: airborne retrieval of $f_{\mathrm{iso}}$. **(b)** As in (a), but for $f_{\mathrm{vol}}$. **(c)** As in (a), but for $f_{\mathrm{geo}}$. **Below: (d)** Corresponding combined scatter plot and histograms between airborne and satellite (resampled on flight track) retrieval for $f_{\mathrm{iso}}$. Error bars correspond to the mean standard deviation of the airborne retrieval. Dashed in scatter plot: 1:1 line. Dashed in histograms: center of axis. **(e)** As in (d), but for $f_{\mathrm{vol}}$. **(f)** As in (d), but for $f_{\mathrm{geo}}$.

## 4.4 Parameterization of snow HDRF

The most significant effects of $\theta_0$, $l_{\mathrm{rough}}$, and $R_{\mathrm{opt}}$ were observed on the model parameter $f_{\mathrm{vol}}$, for which the parameterizations in terms of linear regression fits (black dashed lines in Figs. 6b-8b) are given by:

$$f_{\mathrm{vol}}(\theta_0) = 0.0059 \cdot \theta_0 - 0.0537 \qquad (\text{for } l_{\mathrm{rough}} = 6 - 7\,\mathrm{cm} \text{ and } R_{\mathrm{opt}} = 70 - 85\,\mu\mathrm{m}), \qquad (20)$$

$$f_{\mathrm{vol}}(l_{\mathrm{rough}}) = -0.0656 \cdot l_{\mathrm{rough}}(\mathrm{m}) + 0.3416 \qquad (\text{for } \theta_0 = 55.5° \text{ and } R_{\mathrm{opt}} = 240 - 320\,\mu\mathrm{m}), \qquad (21)$$

$$f_{\mathrm{vol}}(R_{\mathrm{opt}}) = -0.0009 \cdot R_{\mathrm{opt}}(\mu\mathrm{m}) + 0.5013 \qquad (\text{for } \theta_0 = 51.3 - 53.4° \text{ and } l_{\mathrm{rough}} = 4 - 7\,\mathrm{cm}). \qquad (22)$$



The linear correlation coefficients ($R^2$) are 0.98 (for $\theta_0$), 0.75 ($l_{\mathrm{rough}}$), and 0.68 ($R_{\mathrm{opt}}$). Note that these parameterizations are only valid for the wavelength range of about 490-585 nm (green camera channel) and the ranges of all three parameters as given within the descriptions of the case studies.

### 4.5 Comparison with satellite observations

Figure 9 shows the comparison between retrieved $f_k$ from airborne and satellite measurements for the research flight on 29 December 2013, which showed the largest amount of full inversion retrievals for MODIS (235). The flight covered a triangular pattern Southeast of Kohnen station. The top panel shows contour plots with color-coding for the three parameters as retrieved from MODIS on that day: (a) $f_{\mathrm{iso}}$, (b) $f_{\mathrm{vol}}$, and (c) $f_{\mathrm{geo}}$. The colored circles along the flight track (black) correspond to the airborne retrieved $f_k$. The corresponding scatter plots between the MODIS and aircraft retrievals are shown in Figs. 9d-f. The

isotropic and geometric model weights $f_{\mathrm{iso}}$ and $f_{\mathrm{geo}}$ from airborne and MODIS retrievals are in the same order of magnitude and partly agree within the measurement uncertainties. $f_{\mathrm{iso}}$ values from aircraft are slightly higher than the corresponding MODIS retrievals. This is evident from the difference in the PDFs between aircraft and MODIS retrievals as illustrated in Figs. 9d-f. The comparison of $f_{\mathrm{geo}}$ values shows no robust difference between airborne and MODIS retrievals. However, the lowest correlation is found for the volumetric weights $f_{\mathrm{vol}}$. $f_{\mathrm{vol}}$ values retrieved from aircraft range mostly between 0.1 and

0.3 and are much larger than the MODIS retrievals. The map in Fig. 9b shows that the flight covered an area where MODIS retrieved $f_{\mathrm{vol}}$ values of exactly or close to zero. In particular, this holds true for a latitudinal band around 76° S. Even though full inversions were reported for this area, this might still be a measurement artifact. But even for the cases with non-zero values, the aircraft retrievals lead to much higher $f_{\mathrm{vol}}$ values and, thus, larger anisotropy of the surface reflection than shown in the MODIS retrieval.

The combined scatter plots and histograms for the three model weights including all 434 observations from 7 different research flights are shown in Fig. 10 and indicate that the conclusions drawn from the flight on 29 December 2013 are valid for all research flights: $f_{\mathrm{vol}}$ shows the strongest discrepancies between airborne and MODIS retrievals, whereas $f_{\mathrm{iso}}$ values are slightly higher when retrieved from the aircraft measurements. The correlation between airborne and satellite measurements is low for all three model parameters, $R^2$ is 0.18 (for $f_{\mathrm{iso}}$), 0.004 ($f_{\mathrm{vol}}$), and 0.055 ($f_{\mathrm{geo}}$).

Various factors can contribute to the observed differences: (1) data quality, (2) short-term changes in snow properties, (3) footprint size differences, (4) inherent challenges comparing the camera and satellite data.

First, a thorough quality assessment for both datasets formed the prerequisite for the comparison. Only airborne data from cloudless conditions (at solar zenith angles below 70°) with a RMSE lower than 0.1 were compared with full inversion MODIS retrievals (quality flag values 0 or 1, RMSE < 0.1). However, possible artifacts such as the latitudinal band around 76° S with

values of exactly zero cannot be ruled out entirely.

Secondly, the case studies demonstrated the strong influence of the solar zenith angle, the surface roughness, and the optical-equivalent snow grain size on the anisotropy of the reflection. Precipitation events or drifting snow altering the optical-equivalent snow grain size and the surface roughness can change the surface properties on short time scales. These short-term changes in snow properties are not captured well by the 16-day BRDF product from MODIS. Even though the 434 observations





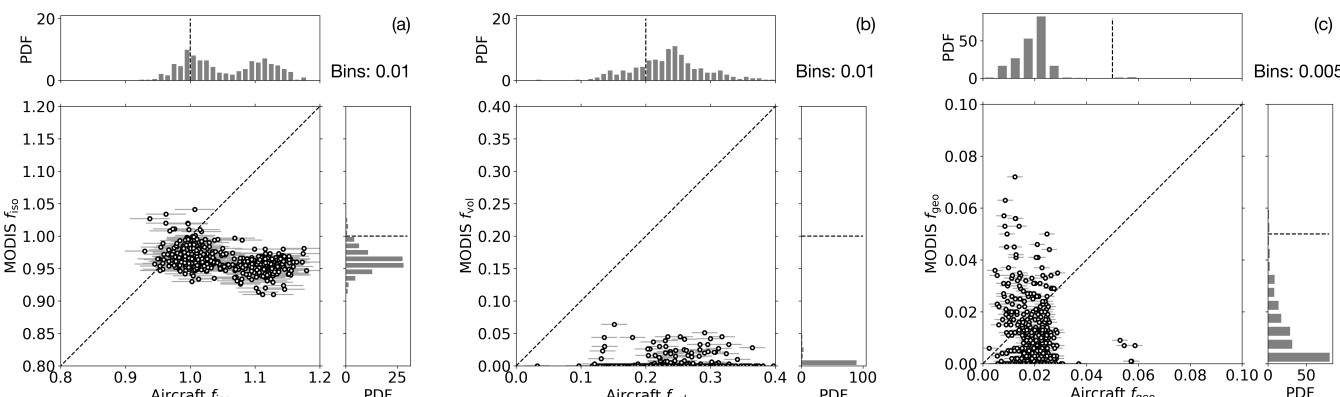

**Figure 10. (a)** Scatter plot between airborne and satellite (resampled on flight track) retrieval for $f_{\mathrm{iso}}$ (490-585 nm wavelength) including all 7 research flights used for the comparison. Error bars correspond to the mean standard deviation of the airborne retrieval. Dashed: 1:1 line. In addition, the histograms for both airborne and satellite retrievals are given. Dashed: center of axis. **(b)** As in (a), but for $f_{\mathrm{vol}}$. **(c)** As in (a), but for $f_{\mathrm{geo}}$.

stem from 7 different research flights, most of the data originates from the two flights on 29 December 2013 and 2 January 2014 (morning flight). For most of the remaining flights, no full inversion retrieval was possible for MODIS. Being restricted to mainly two different days, short-term changes in snow properties can indeed explain the low correlation between the airborne and MODIS retrievals. Also, the solar zenith angle at the time of the aircraft measurements varies from the different zenith

5    angles during the satellite overpasses used for the MODIS retrieval.

Thirdly, the airborne measurements still included varying footprint sizes (mostly between 1 and 3 km) compared to the 500 m resolution of the MODIS product. This was the result of trading off comparability with MODIS and data coverage. Also, no correlation between the $f_k$ retrieved from the camera measurements and the footprint size was found (not shown here).

10    Lastly, comparing the airborne camera data with the MODIS product poses challenges that are inherent to the measurement setup. The observation times of the MODIS product do not coincide with the times of the research flights. MODIS spectral band 4 was chosen as its RSR function agrees best with the green camera channel. As the HDRF is wavelength dependent, differences naturally occur due to different spectral coverage but should be small.

Thus, despite the thorough quality assessment analogously applied to both airborne and satellite data, a quantitative com-
15    parison between the two retrievals seems challenging. Nevertheless, the comparison highlights the difficulties to accurately measure the snow BRDF from MODIS as well as how quickly snow properties can change leading, in this study, to an underestimation of the anisotropic reflection. Many satellite retrievals of surface albedo and atmospheric parameters over snow surfaces rely on a correct angular modeling of the surface reflectance. Thus, inaccuracies in the surface BRDF can propagate and influence other retrieval results.





## 5 Conclusions

The HDRF of snow surfaces in the wavelength band 490-585 nm was investigated with respect to the influence of solar zenith angle, surface roughness, and optical-equivalent snow grain size utilizing an airborne digital 180° fish-eye camera in Antarctica during austral summer in 2013/14. The surface roughness was retrieved using an airborne laser scanner, while the

optical-equivalent snow grain size was calculated from spectral surface albedo measurements. While based at Kohnen station ($75°0'$ S, $0°4'$ E) at the outer part of the East Antarctic Plateau, the airborne measurements with the Polar 6 research aircraft covered an area of around $1000 \times 1000 \,\mathrm{km}^2$ in Dronning Maud Land. Thus, the snow HDRF was obtained for a variety of conditions with different solar zenith angle ($\theta_0$), surface roughness ($l_\mathrm{rough}$), and optical-equivalent snow grain size ($R_\mathrm{opt}$).

The digital camera provides airborne radiance measurements with high angular resolution. The characterization of the sensor

revealed excellent linearity as well as negligible dark current and read-out noise. The HDRF was calculated from the angular radiance measurements applying simulations of the global irradiance with the library for radiative transfer libRadtran using the discrete ordinate radiative transfer solver DISORT. The relative uncertainty of the HDRF measurements is estimated with 4.5 %. The footprint of the snow HDRF measurements analyzed within this study varies between 1 km to 4.5 km.

Three case studies were investigated to separate the effects of $\theta_0$, $l_\mathrm{rough}$, and $R_\mathrm{opt}$ on the snow HDRF. With increasing

solar zenith angle, the HDRF maximum in the forward scattering direction becomes more pronounced. Conversely, roughness structures enhance the backscatter by changing the effective angle of incidence and by casting shadows. This was quantified by inverting the semi-empirical kernel-driven RTLSR model and parameterizing the snow HDRF with respect to $\theta_0$, $l_\mathrm{rough}$, and $R_\mathrm{opt}$ (Eqs. 20-22). The uncertainty of the inversion is estimated with 10 %. The increased anisotropy (with increasing $\theta_0$) is mainly shown by the increase in the model parameter $f_\mathrm{vol}$. Vice versa, $f_\mathrm{vol}$ decreases with increasing $l_\mathrm{rough}$. The snow grain

size reveals a similar effect as surface roughness structures in terms of a decrease in anisotropy with increasing $R_\mathrm{opt}$, which is a different result than found by earlier studies (e.g., Steffen, 1987). Possible reasons have been identified (footprint size, ice crystal shape, contamination with dust particles). However, to yield stronger dependence on the optical-equivalent snow grain size, a camera sensitive to radiation in the near-infrared part of the EM spectrum needs to be employed.

Continuous observations of the snow surface albedo with global coverage can be achieved by satellite observations. However,

the angular modeling of the surface BRDF may introduce large errors especially in polar regions due to low sun elevations and the anisotropic scattering phase function of ice crystals. The RTLSR model was chosen for the inversion as it forms the basis of the MODIS 16-day BRDF/albedo product. Thus, the retrieved model parameters $f_k$ from the HDRF measurements allow for direct comparison with BRDF products from satellite remote sensing. Despite the similar quality assessment applied to airborne and satellite data, large deviations were found especially for the volumetric model weight $f_\mathrm{vol}$. The airborne values

for $f_\mathrm{vol}$ are larger than the corresponding MODIS retrievals. Short-term changes in snow properties (precipitation, drifting snow), which are not captured by the 16-day MODIS retrieval, and different solar zenith angles at the time of measurement are suspected to be the main reasons for the observed differences. Although the effects are presumably small, influences of a varying footprint size for the airborne observations and the different RSR functions of the green camera channel and MODIS spectral band 4 can further contribute to this discrepancies.



Generally, MODIS observations underestimated the anisotropy of the reflection at the snow surfaces. This has possible implications for satellite retrievals of surface albedo as well as atmospheric parameters over snow surfaces (e.g., AOD, cloud properties), which strongly depend on a correct angular modeling of the surface reflectance (e.g., Qu et al., 2015).

The analysis of the snow HDRF measurements can be readily applied to other airborne campaigns. For example, the same

digital camera with a similar setup of the Polar 6 research aircraft was used (a) during the Arctic CLoud Observations Using airborne measurements during polar Day (ACLOUD) between 22 May and 28 June 2017 based at Longyearbyen, Svalbard (Wendisch et al., 2019; Ehrlich et al., 2019; Jäkel and Ehrlich, 2019), and (b) during the Polar Airborne Measurements and Arctic Regional Climate Model Simulation Project (PAMARCMiP) between 12 March and 6 April 2018 at Station North, Greenland (Herber et al., 2012). Thereby, the database of the parameterizations could be extended to wider ranges of $R_{\mathrm{opt}}$,

$l_{\mathrm{rough}}$, and $\theta_0$.

*Data availability.* The tabulated data including the retrieved $f_{\mathrm{iso}}$, $f_{\mathrm{vol}}$, and $f_{\mathrm{geo}}$ as well as $\theta_0$, $l_{\mathrm{rough}}$, $R_{\mathrm{opt}}$, and auxiliary information (time and position of the measurements) will be published for all research flights in the Publishing Network for Geoscientific & Environmental Data (PANGAEA) after submission of this manuscript. The MODIS BRDF/Albedo Model Parameters product (MCD43A1, Version 6) and the quality dataset (MCD43A2) were downloaded from NASA's Land Processes Distributed Active Archive Center (LP DAAC) website

(https://doi.org/10.5067/MODIS/MCD43A1.006 and https://doi.org/10.5067/MODIS/MCD43A2.006).

*Author contributions.* TC performed the post-processing and data analysis of the camera and surface albedo measurements, produced the figures, and wrote the paper with help from all authors. GB and MS conducted the measurements at Kohnen station and with the Polar 6 research aircraft and helped write the paper. VH analyzed the laser scanner data for the surface roughness measurements and helped write the paper. EJ gave substantial input with regard to the data analysis and helped write the paper. AE and MW planned the measurements,

designed the study and helped write the paper.

*Competing interests.* The authors declare that they have no conflict of interest.

*Acknowledgements.* This work was supported by the Deutsche Forschungsgemeinschaft (DFG) in the framework of the priority programme "Antarctic Research with comparative investigations in Arctic ice areas" (SPP 1158) by the grants WE1900/29-1 and BI 816/4-1. We gratefully acknowledge the funding by the Deutsche Forschungsgemeinschaft (DFG, German Research Foundation) – Project-ID 268020496 –

TRR 172, within the Transregional Collaborative Research Center "ArctiC Amplification: Climate Relevant Atmospheric and SurfaCe Processes, and Feedback Mechanisms (AC)[3]". This work was partly supported by the European Research Council (ERC) through Grant StG 758005. We are grateful to the Alfred Wegener Institute, Helmholtz Centre for Polar and Marine Research, Bremerhaven, Germany, for supporting the campaign with logistics, the aircraft and manpower in Antarctica. In addition, we would like to thank Kenn Borek Air Ltd.,





Calgary, Canada, for the great pilots who made the complicated measurements possible. For his insights into the sRaw format of the digital camera, we would like to express our gratitude to Douglas Kerr.



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
