# Peer review of "Parameterizing anisotropic reflectance of snow surfaces from airborne digital camera observations in Antarctica"

_The Cryosphere, 2020_

## Referee Comment (RC1) · Anonymous Referee #1 · 22 Jun 2020

Review of Parameterizing anisotropic reflectance of snow surfaces from airborne digital camera observations in Antarctica

This is a well written paper that examines how well the anisotropic reflectance of snow is parameterized. An important conclusion is that MODIS MCD43 product underestimates the observed anisotropy of snow reflection. Below I discuss my main comments.

1. My comments mostly pertain to how the terms BRDF and HDRF are used. It still remains a bit confusing in your introduction the difference between the BRDF and the HDRF. Since this is still often used non-correctly in many publications it would make sense to be more careful and not use the terms so interchangeably. My understanding

from Schaepman-Strub et al. 2006 is that the BRDF is not something that can be measured whereas the HDRF is. Thus, it is confusing when you then say on line 26 that comparison of in situ measured BRDF with simulations, as the BRDF is not measured in situ as far as I understand it. The equations for each are shown in the methodology, which is helpful, but it does remain a bit confusing they way it is discussed. Since the HDRF is what is actually being measured from the digital camera, I think that all needs to be stated more clearly upfront and it would be good to add discussion on how Eq. 18 relates to Eq. 3 for those less familiar with these topics – i.e. students.

2. I also do not follow follow why the downward irradiance from SMART could not be used to calculate the HDRF. If it's good enough for the surface albedo, and thus the grain size determination, then I do not follow why it's not good enough for the incoming solar radiation. Also, if as you state you are mostly interested in the shape of the HDRF, not the absolute magnitude of the values, then the calibration shouldn't have mattered? I would like to see a comparison between your modeled and measured incoming solar irradiance.

3. What were the cloud vs. clear sky conditions during these flights? Were synoptic cloud observations not also obtained? Could you not see whether or not there were clouds with the SMART measured incoming solar irradiance?

4. I don't follow exactly what was done in section 3.6 for the inversion. How was the HDRF used in this context? Also, seems you setting the HDRF equal to the BRDF in Equation 12 but we don't find that out until section 3.7 so be good to mention it earlier. However, in section 3.7 you then mention you don't expect the atmospheric conditions to be large, which is true though as you mention it is wavelength dependent with the blue channel more impacted. Also the discussion then focuses on the BRF and HDRF relationship with the proportion of direct vs. diffuse but again it's the BRDF that you are substituting with the HDRF so I think it would be better to keep the discussion in that context as the interchanging of terms is hard to follow.

5. It is well known that the anisotropy increases with increasing solar zenith angle so it would be good to reference some early publications that have already discussed this (i.e. some early work by Warren seems relevant here). It's also well known that surface roughness reduces the overall forward scattering as there is more backscatter, so again referencing earlier work is important here. This is also how data from MISR are currently being used to map surface roughness over ice sheets and sea ice.

6. I'm surprised that there is no mention of how BRDF uncertainties translate into albedo and absorbed solar energy uncertainties as that is what is really important after all. While the paper is already quite long, this would complete the study. If the MODIS BRDF model is off, how much does it influence the albedo and the energy balance of the ice sheets?

---

## Referee Comment (RC2) · Anonymous Referee #2 · 3 Jul 2020

The paper describes an approach for characterizing snow directional reflectance using airborne data (spectral surface albedo measurements, and surface roughness determined by means of a laser scanner) that were taken over a large area (1000×1000 km2) located in the East Antarctic Plateau during austral summer in 2013/14 using a single channel (490-585 nm) digital 180° fish-eye camera. The study provides new parameterization of the semi-empirical kernel-driven bidirectional reflectance model (RossThick-LiSparse-Reciprocal (RTLSR)), which is used in the operational MODIS BRDF/albedo product retrieval, based on the airborne data of snow BRDF as a function of solar zenith angle, surface roughness, and optical-equivalent snow grain size. This is an important contribution to snow remote sensing.

[Figure]

General Comments:

1. Section 1: provides an excellent background literature review of snow reflectance anisotropy, snow BRDF models and verification of the models with existing BRDF measurements.

2. Section 2: Methodology – the definitions of reflectance quantities are articulated well, however, the use of symbols is inconsistent. For example, in subsection 2.1, the symbol for the solar zenith angle is given as theta(i) (see Equation 1), then the subscript changes to something else, theta(0) (see Equation 3) without any explanation. In subsection 2.2, the symbol for the relative azimuth angle are different both in Equation 4 and Equation 7.

3. Subsection 3.2 - optical-equivalent grain size retrieval from spectral surface albedo measurements. It was reported elsewhere in the manuscript (subsection 3.4) that "the downward irradiance measurements from SMART could not be used for the calculation of the HDRF due to calibration issues." So does this imply that the optical-equivalent snow grain size retrieved from SMART measurements may be affected by those calibration issues? It's also odd that the "optical-equivalent snow grain size from SMART and analogous ground-based measurements were validated against grain size observations utilizing reflectance measurements with MODIS." Which retrievals are assumed to be the "truth"?

4. Subsection 3.3.2 – Radiance calibration and image post-processing. This subsection provides important details about the post processing, but there are no details about the pre-deployment calibration. Given that the flights were performed between 24 December 2013 and 5 January 2014, it may be shown that each flight need a different calibration, which would depend on the calibration stability of the digital camera Canon EOS-1D Mark III.

Specific comments: 1. Pg. 1. Line 17: The study finds that "MODIS observations generally underestimated the anisotropy of the surface reflection", but it would help to

provide the degree to which MODIS underestimates the anisotropy of surface reflection.

2. Pg. 2, Line 4: The statement, "Satellites monitor the reflectance (i.e., reflected radiance in units of W m−2 sr−1) at the top of the atmosphere (TOA). However, they are restricted in terms of the number of available observation angles and spectral bands as well as their temporal resolution." Is this in reference to polar orbiting satellites or geostationary satellites?

3. Pg. 2, Line 8, the statement, "During the first step, the TOA reflectance is converted into a surface reflectance by means of an atmospheric radiative transfer parameterization," need to be clarified. What is "atmospheric radiative transfer parameterization?"

4. Pg. 2, line 10, the recommended terminology: bidirectional reflectance-distribution function (BRDF). Note the "reflectance-distribution" is a compound adjective and should be hyphenated.

5. Pg. 3, line 19, change the statement, "Comparing this asymptotic model to in situ observations of the BRDF, …" to "Evaluating this asymptotic model with in situ observations of the BRDF, …"

6. Pg. 3, in the paragraph, marked by lines 8-25, various snow reflectance terms are mentioned: snow BRDF (line 10), a polarized BRDF (line 12), reflectance of snow (line 18), inherent optical properties (line 22), and bidirectional reflectance (line 24). This may be confusing to readers and it would help to clarify or use consistent terms.

7. Pg. 3, line 29, the use of "solar azimuth" and "azimuthal directions" in this sentence is confusing, "Kuhn (1985) observed a peak in reflectance in the azimuthal directions up to 60° to both sides of the solar azimuth that becomes more prominent with increasing solar zenith angle and snow grain size." What is the zero azimuth in this case?

8. Pg. 4, line 28, change "480" to "340". The smallest CAR wavelength is 340 nm.

9. Pg. 5, line 29, the BRDF symbol used here is not consistent with Schaepman-Strub
et al. (2006) as stated/line 27.

10. Pg. 5, line 30, the statement "the reflected radiance for all reflection angles . . ." is unclear. The entire definition of the spectral BRDF need to be clarified.

11. Pg. 13, line 14, the location of each pixel is given by x and y, but "x" and "y" are not defined elsewhere in the manuscript.

12. Pg. 6, the derivation of Equation (3) is not clear.

13. Pg. 13, line 19, the author need to show how " . . . the viewing angles are corrected" using the "the roll, pitch and yaw angles of the aircraft at the time of measurement."

14. Pg. 13, Equation 18, the viewing and illumination geometry variables need to be defined taking into account the aircraft orientation. The downward flux F should also be described here immediately after the equation is listed, unless it's previously defined.

―――――――――――――――

---

## Short Comment (SC1) · 20 Jul 2020

Dear Authors,

I have a little note. Kokhanovsky and Zege (2004) do not represent the snow grains as fractal particles. They consider any particles with no regard to their shape.

Truly yours, Aleksey Malinka.

---

## Author Comment (AC1) · 31 Aug 2020

**Replies to comments of Anonymous Referee #1**

1. My comments mostly pertain to how the terms BRDF and HDRF are used. It still remains a bit confusing in your introduction the difference between the BRDF and the HDRF. Since this is still often used non-correctly in many publications it would make sense to be more careful and not use the terms so interchangeably. My understanding from Schaepman-Strub et al. 2006 is that the BRDF is not something that can be measured whereas the HDRF is. Thus, it is confusing when you then say on line 26 that comparison of in situ measured BRDF with simulations, as the BRDF is not measured in situ as far as I understand it. The equations for each are shown in the methodology, which is helpful, but it does remain a bit confusing they way it is discussed. Since the HDRF is what is actually being measured from the digital camera, I think that all needs to be stated more clearly upfront and it would be good to add discussion on how Eq.18 relates to Eq. 3 for those less familiar with these topics – i.e. students.

Thank you for this comment. We agree that the used terminology and the differentiation between HDRF and BRDF is done too sloppily in some other publications. We tried to be careful in this matter, however as you correctly pointed out, some inconsistencies slipped our attention. In case the actual measured quantity was the HDRF, we changed the term BRDF to HDRF

5 (although stated differently in the respective publication). We also define an 'effective BRDF' as done in Gatebe and King (2016) in case the FOV is small and the atmospheric influence is considered. We believe this is reasonable for measurements with the CAR instrument and decided to reference the CAR database in the same way as reported by Gatebe and King (2016) as this constitutes an important dataset.

We agree that, so far, the relation between Eq. 18 and Eq. 3 was not obvious. We therefore omit the rather theoretical definition
given so far in the manuscript (Eq. 3) and give already the more conceptual definition via the BRF. This way, Eq. 3 already has the form of Eq. 18 and the relation should become more clear to the reader.

**Changes in text:**

- Page 3 Line 29: The comparison of in situ measurements of the snow reflectance with simulations is essential in terms of
- 15

20

model validation. Observations of the HDRF (or effective BRDF in case the FOV is small and the atmospheric influence is considered) are conducted using a variety of different measurement concepts.

- P4 L10: 'Several studies observed systematically less anisotropy for a typical snow BRDF than estimated from simulations [...]': we replaced 'BRDF' with 'HDRF'
- P4 L19: 'Thus, the latter are more suitable for studying the influence of macroscopic surface roughness on the surface BRDF.': we replaced 'BRDF' with 'HDRF'
- P4 L26: 'and the snow BRDF loses its azimuthal symmetry (Warren et al., 1998)': we replaced 'BRDF' with 'HDRF'
- P5 L8: 'Cox and Munk (1954) analyzed radiance calibrated analog photographs for the parameterization of the ocean BRDF': we replaced 'BRDF' with 'HDRF'

- P5 L10: 'The instantaneous measurement of multiple viewing angles facilitates aerial BRDF measurements with digital cameras.': we replaced 'BRDF' with 'HDRF'
- P6 L5: As the BRDF of an ideal Lambertian surface is  $(\pi sr)^{-1}$ , the BRF is given by

$$R_{\rm BRF} = \frac{\mathrm{d}I_{\rm r}\left(\theta_{\rm i},\varphi_{\rm i};\theta_{\rm r},\varphi_{\rm r}\right)}{\mathrm{d}F_{\rm i}\left(\theta_{\rm i},\varphi_{\rm i}\right)} \cdot \frac{\mathrm{d}F_{\rm i}\left(\theta_{\rm i},\varphi_{\rm i}\right)}{\mathrm{d}I_{\rm r}^{\rm ideal}\left(\theta_{\rm i},\varphi_{\rm i}\right)} = \pi\,\mathrm{sr}\cdot f_{\rm BRDF}.$$

5

In Eq. 2 and in the remainder of this section, the spectral dependence is omitted for reasons of simplicity. In atmospheric conditions, both the BRDF and BRF cannot be measured directly as the global irradiance  $F_0$  reaching the surface is composed of a direct ( $F_i$ ) and diffuse ( $F_{diff}$ ) component. In this case, the measurable quantity is the HDRF. The definition of the HDRF is analogous to the BRF, but includes irradiance from the entire hemisphere (denoted with  $2\pi$ ):

$$\begin{split} R_{\rm HDRF} &= \pi \, {\rm sr} \cdot \frac{{\rm d} I_{\rm r} \left(\theta_{\rm i}, \varphi_{\rm i}, 2\pi; \theta_{\rm r}, \varphi_{\rm r}\right)}{{\rm d} F_0 \left(\theta_{\rm i}, \varphi_{\rm i}, 2\pi\right)} \\ &= R_{\rm BRF} (\theta_{\rm i}, \varphi_{\rm i}; \theta_{\rm r}, \varphi_{\rm r}) \cdot f_{\rm dir} + R(2\pi; \theta_{\rm r}, \varphi_{\rm r}) \cdot (1 - f_{\rm dir}) \,. \end{split}$$

10 – new Eq. 18:

$$R_{\rm HDRF}(\theta_0,\varphi_0;\theta_{\rm r},\varphi_{\rm r}) = \frac{\pi \operatorname{sr} \cdot I(\theta_{\rm r},\varphi_{\rm r})}{F^{\downarrow}(\theta_0,\varphi_0)}$$

2. I also do not follow why the downward irradiance from SMART could not be used to calculate the HDRF. If it's good enough for the surface albedo, and thus the grain size determination, then I do not follow why it's not good enough for the incoming solar radiation. Also, if as you state you are mostly interested in the shape of the HDRF, not the absolute magnitude of the values, then the calibration shouldn't have mattered? I would like to see a comparison between your modeled and measured incoming solar irradiance.

The grain size determination relies on surface albedo measurements. The calculation of the surface albedo from the raw spectra measured with SMART involves three main steps: (a) the correction for dark and stray light, (b) the correction for the non-ideal angular response of the sensor–spectrometer system, and (c) the cross- calibration of the up- and downward looking optical

15 inlets to account for different sensitivities of the sensors-spectrometer systems. The latter part (c) was performed in the field during the observations and comprised successive measurements of the signals of the up- and downward-looking sensors under the same illumination with an integrating sphere. Thus, for albedo measurements, an absolute calibration converting the digital numbers registered by the spectrometers into units of irradiance is not required. All calibration steps for measurements of the surface albedo could be carried out, which is why the retrieval of the optical-equivalent snow grain size from SMART
20 measurements is not affected by these calibration issues.

This is different for the downward irradiance that requires an absolute calibration converting the raw signal into units of irradiance. The absolute calibration was performed in the laboratory. Normally, the cross-calibration is used to transfer the absolute calibration as measured in the laboratory to the field to account for changes in the transmissivity of the sensor–fiber–spectrometer system. Unfortunately, problems with the power supply of the integrating sphere in the field occured,

and the integrating sphere got destroyed after the campaign before it could be used later for an additional laboratory calibration of SMART. Due to this failed transfer calibration, an absolute calibration was not possible and the downward irradiance measurements from SMART could not be used for the calculation of the HDRF. For the same reason, a comparison between modeled and measured incoming solar irradiance is not possible.

- Instead, the global irradiance was simulated along the flight track with libRadtran using DISORT. The simulated irradiance 5 was integrated over the wavelength range of each camera channel and weighted with the RSR function of the camera. The use of simulations limits the validity of absolute values of the measured HDRF to cloudless conditions. It is true that we are mostly interested in the shape of the HDRF. However, the retrieved values for the model parameter  $f_{iso}$  would be influenced by the absolute values of the HDRF. We therefore restrict our analysis to cloudless cases when we are confident that the simulated
- 10 values for the downward irradiance were representative of the actual measurement conditions.

As both reviewers mentioned this, we added a clarification in the manuscript.

**Changes in text:**

15 - P14 L4: The downward irradiance measurements from SMART could not be used for the calculation of the HDRF due to calibration issues. Note that this pertains to the radiometric calibration only, which converts the digital numbers registered by the spectrometer into units of irradiance. For the albedo measurements with SMART, a relative calibration of the upper and lower sensors is sufficient and an absolute radiometric calibration is not required. Thus, the albedo measurements and the retrieval of the optical-equivalent snow grain size are unaffected by this calibration issues. For the 20 calculation of the HDRF, the global irradiance was simulated [...] instead.

3. What were the cloud vs. clear sky conditions during these flights? Were synoptic cloud observations not also obtained? Could you not see whether or not there were clouds with the SMART measured incoming solar irradiance?

In approximately 75 % of the camera observations, cloudless conditions prevailed. Indeed, cloudless cases were identified from visual synoptic observations during the flights as well as using the downward measured irradiance by SMART. Periods with fast fluctuations in the downward irradiance were flagged as cloudy. Clouds influence the HDRF measurements in changing the direct/diffuse ratio of incoming solar radiation. For a quantitative analysis, the cloud scene during the times of measurement (cloud cover, optical thickness, etc.) needs to be well characterized, which is not possible from the observations available during the flights. Thus, we restricted our analysis to cloudless conditions only.

- **Changes in text:**
- P14 L13: The use of simulations limits the validity of absolute values of the measured HDRF to cloudless conditions. Cloudless cases (approximately 75% of the camera observations) were identified from visual synoptic observations

30

25

during the flights as well as using the downward irradiance measured with SMART. Periods with fast fluctuations in the downward irradiance were flagged as cloudy. Although mainly the shape of the HDRF is analyzed within this work (which is independent from the absolute value of  $F^{\downarrow}$ ), the analysis is restricted to cloudless conditions only.

4. I don't follow exactly what was done in section 3.6 for the inversion. How was the HDRF used in this context? Also, seems you setting the HDRF equal to the BRDF in Equation 12 but we don't find that out until section 3.7 so be good to mention it earlier. However, in section 3.7 you then mention you don't expect the atmospheric conditions to be large, which is true though as you mention it is wavelength dependent with the blue channel more impacted. Also the discussion then focuses on the BRF and HDRF relationship with the proportion of direct vs. diffuse but again it's the BRDF that you are substituting with the HDRF so I think it would be better to keep the discussion in that context as the interchanging of terms is hard to follow.

Thank you for suggesting this improved structure of the manuscript. We agree that it is helpful for the reader to mention the substitution of the BRDF with the HDRF in Equation 12 earlier. We therefore changed the structure as follows: we switched Sects.
3.6 and 3.7 and added the suggested clarification at the beginning of the new Sect. 3.6. Concerning your second suggestion, we believe it is better to continue discussing the BRF and HDRF relationship as this is what was simulated by Schaepman-Strub et al. (2006). However, we added a reminder about the close relation between the BRF and BRDF by referring to Eq. 2 at this point.

10

**Changes in text:**

- P15 L2: For each averaged HDRF, the weighting functions  $f_{iso}$ ,  $f_{vol}$ , and  $f_{geo}$  are retrieved by inverting the RTLSR BRDF model and setting the HDRF equal to the BRDF in Eq. 12. This is necessary as the BRDF is not measurable under atmospheric conditions. However, the atmospheric influence [...]

15

- P15 L11: Schaepman-Strub et al. (2006) simulated the difference between the HDRF and BRF ( $R_{BRF} = \pi \cdot f_{BRDF}$ , compare Eq. 2) for snow surfaces

- switched Section 3.6 and 3.7

5. It is well known that the anisotropy increases with increasing solar zenith angle so it would be good to reference some early publications that have already discussed this (i.e. some early work by Warren seems relevant here). It's also well known that surface roughness reduces the overall forward scattering as there is more backscatter, so again referencing earlier work is important here. This is also how data from MISR are currently being used to map surface roughness over ice sheets and sea ice.

We totally agree and added more references in the discussion part of the manuscript.

**Changes in text:**

- P19 L18: This expected increase in the anisotropy of the snow HDRF for increasing θ0 is obvious in the changing model parameters fk and has been discussed in earlier studies (e.g., Dirmhirn and Eaton, 1975; Kuhn, 1985; Warren et al., 1998; Hudson et al., 2006).
- P20 L1: This is in accordance with Warren et al. (1998) who observed a reduction of the forward reflection peak due to sastrugi during tower measurements at South Pole Station. The effect of surface roughness on the BRDF has been previously investigated in observational (e.g., Grenfell et al., 1994; Hudson and Warren, 2007) and modeling studies (e.g., Leroux and Fily, 1998; Zhuraleva and Kokhanovsky, 2011) and is facilitated by the multi-angular reflectance data from MISR to map surface roughness over ice sheets and sea ice from space (Nolin et al., 2002).

6. I'm surprised that there is no mention of how BRDF uncertainties translate into albedo and absorbed solar energy uncertainties as that is what is really important after all. While the paper is already quite long, this would complete the study. If the MODIS BRDF model is off, how much does it influence the albedo and the energy balance of the ice sheets?

- 10 Thank you for this suggestion. We agree that the influence on the surface energy budget would be an important point. While it would be possible to translate the retrieved model parameters into a spectral albedo corresponding to the wavelength range of the camera channel (490-585 nm for green), it is the broadband albedo that controls the surface energy budget. For the narrow-to-broadband albedo conversion, the MODIS albedo product uses precomputed coefficients and information from all spectral channels of MODIS. The camera observations could ideally only cover a spectral range between 400 to 700 nm (compare Figure 3). As the HDRF is strongly wavelength dependent (e.g., Hudson et al., 2006; Marks et al., 2015; Gatebe and King,
- 1000 nm where ice absorption becomes dominant. All in all, these points would make an attempt at a surface energy budget calculation doubtful in our view.

20

---

## Author Comment (AC2) · 31 Aug 2020

**Replies to comments of Anonymous Referee #2**

General Comments

1. Section 1: provides an excellent background literature review of snow reflectance anisotropy, snow BRDF models and verification of the models with existing BRDF measurements.

Thank you! We tried to be more rigorous in distinguishing between the BRDF and HDRF as some inconsistencies in referencing other studies slipped our attention. In case the actual measured quantity was the HDRF, we changed the term BRDF to HDRF (although stated differently in the respective publication). We further added a publication from an earlier study using airborne camera data for BRDF observations for forest classification.

**Changes in text:**

 – Koukal, T. and Atzberger, C.: Potential of multi-angular data derived from a digital aerial frame camera for forest classification, IEEE J. Sel. Topics Appl. Earth Observ. Remote Sens., 5, 30–43, https://doi.org/10.1109/JSTARS.2012.2184527, 2012.

2. Section 2: Methodology – the definitions of reflectance quantities are articulated well, however, the use of symbols is inconsistent. For example, in subsection 2.1, the symbol for the solar zenith angle is given as theta(i) (see Equation 1), then the subscript changes to something else, theta(0) (see Equation 3) without any explanation. In subsection 2.2, the symbol for the relative azimuth angle are different both in Equation 4 and Equation 7.

Thanks a lot for spotting these typos. We introduce the reflectance quantities and the modeling (Sects. 2.1 and 2.2) with the general illumination zenith and azimuth angles ($\theta_i$ and $\varphi_i$) as, in principle, the illumination source does not need to be the Sun. Only when referring to specific measurements (from Sect. 3 on), we use $\theta_0$ and $\varphi_0$ as the solar zenith and azimuth angle, respectively. We changed the subscripts in Eqs. 3-11 accordingly and use the consistent symbol $\Delta\varphi$ for the relative azimuth angle in Eqs. 4-11.

**Changes in text:**

 – Changed subscripts in Eqs. 3-11.

3. Subsection 3.2 - optical-equivalent grain size retrieval from spectral surface albedo measurements. It was reported elsewhere in the manuscript (subsection 3.4) that "the downward irradiance measurements from SMART could not be used for the calculation of the HDRF due to calibration issues." So does this imply that the optical-equivalent snow grain size retrieved from SMART measurements may be affected by those calibration issues? It's also odd that the "optical-equivalent snow grain size from SMART and analogous ground-based measurements were validated against grain size observations utilizing reflectance measurements with MODIS." Which retrievals are assumed to be the "truth"?

All calibration steps for measurements of the surface albedo could be carried out, which is why the retrieval of the optical-equivalent snow grain size from SMART measurements is not affected by these calibration issues. This is different for the downward irradiance that requires an absolute radiometric calibration converting the raw signal into units of irradiance. The calibration issues were related to the radiometric calibration only. Thus, the downward irradiance measurements from SMART could not be used for the calculation of the HDRF. The other reviewer asked the same question, for more details please see our answer to Comment #2 of Reviewer #1.

As both reviewers mentioned this, we added a clarification in the manuscript.

Thank you also for the second part of your comment. It was not our intention to assume one retrieval to be the truth. We rephrased the sentence accordingly.

**Changes in text:**

– Page 14 Line 4: The downward irradiance measurements from SMART could not be used for the calculation of the HDRF due to calibration issues. Note that this pertains to the radiometric calibration only, which converts the digital numbers registered by the spectrometer into units of irradiance. For the albedo measurements with SMART, a relative calibration of the upper and lower sensors is sufficient and an absolute radiometric calibration is not required. Thus, the albedo measurements and the retrieval of the optical-equivalent snow grain size are unaffected by this calibration issues. For the calculation of the HDRF, the global irradiance was simulated [...] instead.

– P10 L10: The retrieved $R_\mathrm{opt}$ from SMART and analogous ground-based measurements were compared to grain size observations utilizing reflectance measurements with MODIS (Carlsen et al., 2017).
* * *
4. Subsection 3.3.2 – Radiance calibration and image post-processing. This subsection provides important details about the post processing, but there are no details about the pre-deployment calibration. Given that the flights were performed between 24 December 2013 and 5 January 2014, it may be shown that each flight need a different calibration, which would depend on the calibration stability of the digital camera Canon EOS-1D Mark III.
* * *
Thank you for your comment. Due to the problems with the integrating sphere during the campaign (compare Comment #2 from Reviewer #1), no transfer calibration in the field could be analyzed. However, unlike for SMART, there is no reason to expect a change of the camera calibration as no mechanical parts are unplugged during the deployment (laboratory vs. field). A comparison of laboratory calibrations before and after the campaign and for different temperature regimes did not show significant changes for the camera calibration.

Specific Comments
* * *
1. Pg. 1. Line 17: The study finds that "MODIS observations generally underestimated the anisotropy of the surface reflection", but it would help to provide the degree to which MODIS underestimates the anisotropy of surface reflection.
* * *
Suggestion taken, thank you. We added the average factor of underestimation for the volumetric model weight (which is the dominant contribution to the anisotropy in the analyzed cases). The factor 10 was calculated when neglecting the possible artifacts such as the latitudinal band around 76° S with MODIS values for $f_\mathrm{vol}$ of exactly zero. We added this information in the abstract, results and discussion, and conclusions.

**Changes in text:**

- P1 L16: For the analyzed cases, MODIS observations (545-565 nm wavelength band) generally un-derestimated the anisotropy of the surface reflection. The largest deviations were found for the vol-umetric model weight $f_{\mathrm{vol}}$ (average underestimation by factor of 10).

- P24 L32: For the analyzed cases, MODIS underestimates the volumetric model weight $f_{\mathrm{vol}}$ on aver-age by at least a factor of 10 compared to the airborne measurements (neglecting MODIS values of exactly zero).

- P26 L10: The airborne values for $f_{\mathrm{vol}}$ are larger than the corresponding MODIS retrievals (by at least a factor of 10).

2. Pg. 2, Line 4: The statement, "Satellites monitor the reflectance (i.e., reflected radiance in units of $\mathrm{Wm}^{-2}\mathrm{sr}^{-1}$) at the top of the atmosphere (TOA). However, they are restricted in terms of the number of available observation angles and spectral bands as well as their temporal resolution." Is this in reference to polar orbiting satellites or geostationary satellites?

Thank you for this question. The statement is in reference to polar-orbiting satellites such as relevant for the MODIS instrument onboard Aqua/Terra satellites. The temporal resolution for geostationary satellites is of course much better, however the use of geostationary satellites for global surface albedo measure-ments comes with the drawback of less global coverage (especially missing polar regions) and the need to generate consistent estimates from different sensors. Eventually, this leads to different algorithms for surface albedo estimates for polar-orbiting and geostationary satellites. We clarified the relevance for polar-orbiting satellites in the manuscript.

**Changes in text:**

- P2 L4: Polar-orbiting satellites monitor the reflectance [...]

- P2 L6: The processing of measurements from polar-orbiting satellites for monitoring the broadband surface albedo [...]

3. Pg. 2, Line 8, the statement, "During the first step, the TOA reflectance is converted into a surface reflectance by means of an atmospheric radiative transfer parameterization," need to be clarified. What is "atmospheric radiative transfer parameterization?"

With 'atmospheric radiative transfer parameterization' we mean that the atmospheric correction algorithm has to correct for gaseous and aerosol scattering and absorption and surface-atmosphere-coupling effects (e.g. due to the anisotropic BRDF of the surface) using radiative transfer modeling. To avoid confusion in that regard, we clarified the statement in the manuscript. Thanks for this comment.

**Changes in text:**

- P2 L8: During the first step, the TOA reflectance is converted into a surface reflectance by correcting for gaseous and aerosol scattering and absorption applying radiative transfer modeling (e.g., Vermote and Kotchenova, 2008).

4. Pg. 2, line 10, the recommended terminology: bidirectional reflectance-distribution function (BRDF). Note the "reflectance-distribution" is a compound adjective and should be hyphenated.

10 Thank you for pointing this out. As this might be correct, we prefer omitting the additional hyphen for reasons of consistency applying the terminology as defined in Schaepman-Strub et al. (2006).

5. Pg. 3, line 19, change the statement, "Comparing this asymptotic model to in situ observations of the BRDF,..." to "Evaluating this asymptotic model with in situ observations of the BRDF,..."

Changed as suggested. To be more consistent with regard to the impossibility to measure the BRDF in situ, we also rephrased it to 'in situ measurements of snow reflectance'.

15

**Changes in text:**

11

- P3 L21: Evaluating this asymptotic model with in situ measurements of the snow reflectance, Kokhanovsky et al. (2005) found a generally good agreement but a reduced model accuracy in the solar principal plane at large observation angles.

6. Pg. 3, in the paragraph, marked by lines 8-25, various snow reflectance terms are mentioned: snow BRDF (line 10), a polarized BRDF (line 12), reflectance of snow (line18), inherent optical properties (line 22), and bidirectional reflectance (line 24). This may be confusing to readers and it would help to clarify or use consistent terms.

Thanks for pointing this out. We rephrased, where possible, some of the terms in order to make it easier for the reader. However, the terms mentioned in the paragraph are not all interchangeable and need to remain as stated in order to stay rigorous. We added some details for 'inherent optical properties' for clarity. The bidirectional reflectance (former line 24) cannot be replaced, as Malinka (2016) calculate the BRF which is not yet defined in the manuscript and we prefer to leave it like that to avoid confusion.

**Changes in text:**

- P3 L14: changed 'polarized BRDF model' to 'snow BRDF model including polarization'

- P3 L20: changed 'to model the reflectance of snow' to 'to model the BRDF of snow'

- P3 L24: added details for 'inherent optical properties': 'inherent optical properties (extinction coefficient, single scattering albedo, scattering phase function, and the polarization properties)'

7. Pg. 3, line 29, the use of "solar azimuth" and "azimuthal directions" in this sentence is confusing, "Kuhn (1985) observed a peak in reflectance in the azimuthal directions up to 60° to both sides of the solar azimuth that becomes more prominent with increasing solar zenith angle and snow grain size." What is the zero azimuth in this case?

To avoid a possible confusion, we rephrased this sentence. The zero azimuth is the solar azimuth angle, the $\pm 60°$ refer to the azimuth angle range of reflected radiation relative to the solar azimuth angle.

**Changes in text:**

- P3 L32: Kuhn (1985) observed a peak in reflectance that is contained within $\pm 60°$ to both sides of the solar azimuth angle. This peak becomes more prominent with increasing solar zenith angle and snow grain size.

8. Pg. 4, line 28, change "480" to "340". The smallest CAR wavelength is 340 nm.

Thank you for spotting this. We updated the CAR wavelength limits to 340 and 2324 nm and reference the earlier Gatebe et al. (2003) publication (instead of Gatebe et al., 2005).

**Changes in text:**

- P4 L30: The effective BRDFs were acquired by the Cloud Absorption Radiometer (CAR, Gatebe et al. 2003) over a 30-year period between 1984 and 2014. The CAR is a scanning radiometer covering 14 spectral channels between 340 and 2324 nm

- Gatebe, C., King, M., Platnick, S., Arnold, G., Vermote, E., and Schmid, B.: Airborne spectral measurements of surface-atmosphere anisotropy for several surfaces and ecosystems over southern Africa, J. Geophys. Res., 108, 8489, https://doi.org/doi:10.1029/2002JD002397, 2003.

9. Pg. 5, line 29, the BRDF symbol used here is not consistent with Schaepman-Strub et al. (2006) as stated/line 27.

Thank you for this comment. It is true that Schaepman-Strub et al. (2006) denote the BRDF with $f_{\mathrm{r}}(\theta_{\mathrm{i}}, \varphi_{\mathrm{i}}; \theta_{\mathrm{r}}, \varphi_{\mathrm{r}})$, the BRF with $R(\theta_{\mathrm{i}}, \varphi_{\mathrm{i}}; \theta_{\mathrm{r}}, \varphi_{\mathrm{r}})$, and the HDRF with $R(\theta_{\mathrm{i}}, \varphi_{\mathrm{i}}, 2\pi; \theta_{\mathrm{r}}, \varphi_{\mathrm{r}})$. To better distinguish between the reflectance functions $f()$ (in units $\mathrm{sr}^{-1}$) and the dimensionless reflectance factors $R$, we changed the notation for the BRF and HDRF to $R_{\mathrm{BRF}}$ and $R_{\mathrm{HDRF}}$. This is more consistent with Schaepman-Strub et al. (2006), however we decided to keep the acronyms in the indices of the reflectance factors as the difference in notation between the BRF ($R(\theta_{\mathrm{i}}, \varphi_{\mathrm{i}}; \theta_{\mathrm{r}}, \varphi_{\mathrm{r}})$) and HDRF ($R(\theta_{\mathrm{i}}, \varphi_{\mathrm{i}}, 2\pi; \theta_{\mathrm{r}}, \varphi_{\mathrm{r}})$) is only marginal and hard to identify, especially for someone not familiar with the details.

**Changes in text:**

– We changed the notation throughout the manuscript from $f_{\mathrm{BRF}}$ and $f_{\mathrm{HDRF}}$ to $R_{\mathrm{BRF}}$ and $R_{\mathrm{HDRF}}$, respectively.

10. Pg. 5, line 30, the statement "the reflected radiance for all reflection angles..." is unclear. The entire definition of the spectral BRDF need to be clarified.

Thanks, we clarified the definition of the spectral BRDF by including the definition of the direction of reflection.

**Changes in text:**

– P5 L33: The spectral BRDF ($f_{\mathrm{BRDF}}$, unit of $\mathrm{sr}^{-1}$) provides for each solar zenith ($\theta$) and azimuth angle ($\varphi$) of incident direct irradiance $F_{\mathrm{i}}(\theta_{\mathrm{i}}, \varphi_{\mathrm{i}}; \lambda)$ the reflected radiance $I_{\mathrm{r}}(\theta_{\mathrm{i}}, \varphi_{\mathrm{i}}; \theta_{\mathrm{r}}, \varphi_{\mathrm{r}}; \lambda)$ for all directions of reflection (defined by the reflection zenith and azimuth angles $\theta_{\mathrm{r}}$ and $\varphi_{\mathrm{r}}$) by [...]

11. Pg. 13, line 14, the location of each pixel is given by x and y, but "x" and "y" are not defined elsewhere in the manuscript.

10  $x$ and $y$ define the location of each pixel in terms of rows and columns on the sensor. We added this detail when first defining $x$ and $y$ in Sect. 3.3.2 right before Equation (16).

**Changes in text:**

– P13 L6: The calibration factor $k_{\mathrm{c}}$ is defined at each pixel location on the sensor (row $x$, column $y$) as
15    [...]

12. Pg. 6, the derivation of Equation (3) is not clear.

Thanks for pointing this out. However, the derivation of Equation (3) is lengthy and beyond the scope of this manuscript. We refer to Schaepman-Strub et al. (2006) for details.

**Changes in text:**

– P6 L13: The second step in Eq. 3 assumes that the incident diffuse radiation is isotropic (for details, see Schaepman-Strub et al., 2006).

13. Pg. 13, line 19, the author need to show how "...the viewing angles are corrected" using the "the roll, pitch and yaw angles of the aircraft at the time of measurement."

So far, the correction of the viewing angles was explained earlier in the text. However, to avoid this potential confusion, we decided to provide all information about the aircraft attitude correction in one paragraph.

**Changes in text:**

– P13 L25: The camera viewing angles are calculated from the geometric calibration for each pixel ($x$, $y$). As the camera is fixed to the aircraft frame, a correction for the aircraft attitude was implemented to obtain the reflection angles $\theta_\mathrm{r}$ and $\varphi_\mathrm{r}$. Thus, beside the geometric calibration, the observed reflection angles are determined by the attitude angles of the aircraft. Utilizing the data from the internal navigation system (INS) and the global positioning system (GPS) on Polar 6, the viewing angles are corrected depending on the roll, pitch and yaw angles of the aircraft at the time of measurement. In this regard, Euler rotations are applied as described in Ehrlich et al. (2012).

14. Pg. 13, Equation 18, the viewing and illumination geometry variables need to be defined taking into account the aircraft orientation. The downward flux F should also be described here immediately after the equation is listed, unless it's previously defined.

Thank you for this comment. The reflection geometry is obtained from the viewing geometry by means of the aircraft attitude correction. We believe in moving the aircraft attitude correction paragraph to directly above Eq. 18 (see comment #13 above), the illumination and reflection geometry are now sufficiently defined. We added the illumination angles $\theta_0$ and $\varphi_0$ to Eq. 18. The downward flux $F^\downarrow$ is already defined in Sect. 3.2, but we added a description for clarity.

**Changes in text:**

15

– P13 L30: Finally, the observed HDRF is calculated by:

$$R_{\mathrm{HDRF}}(\theta_0, \varphi_0; \theta_{\mathrm{r}}, \varphi_{\mathrm{r}}) = \frac{\pi \, \mathrm{sr} \cdot I(\theta_{\mathrm{r}}, \varphi_{\mathrm{r}})}{F^{\downarrow}(\theta_0, \varphi_0)} \,.$$

$F^{\downarrow}(\theta_0, \varphi_0)$ denotes the downward solar irradiance at the time of measurement (with solar zenith and azimuth angles $\theta_0$ and $\varphi_0$).

---

## Author Comment (AC3) · 31 Aug 2020

**Reply to short comment by Aleksey Malinka**

Dear Authors, I have a little note. Kokhanovsky and Zege (2004) do not represent the snow grains as fractal particles. They consider any particles with no regard to their shape. Truly yours, Aleksey Malinka.

Dear Alekesey Malinka,

Thank you for your interest in our manuscript and your efforts to improve it. We checked Kokhanovsky
and Zege (2004) again following your comment: they indeed checked different particle models for the phase function of ice crystals, namely the spherical model, fractal model, and random particle model (which we believe you are referring to in your comment). However, in the last paragraph on page 5, they state: "We choose the fractal model here because it has no free parameters for nonabsorbing ice particles and corresponds closely to the random-particle model at extreme values of its randomness pa-
rameters." Kokhanovsky et al. (2005) compare the developed approximate asymptotic theory with in situ measurements. On page 2 they state: "Kokhanovsky and Zege (2004) derived the following relationship [...] in the framework of the approximation of fractal snow grains." Therefore, we are confident that the Kokhanovsky and Zege (2004) work is referenced correctly in our Introduction. But thanks again for looking at this and encouraging us to double-check.

Kind regards from the authors.